# The role of training variability for model-based and model-free learning of an arbitrary visuomotor mapping

**Carlos A. Velázquez-Vargas** [1]*, **Nathaniel D. Daw**[1,2], **Jordan A. Taylor**[1,2]

**1** Department of Psychology, Princeton University, Princeton, New Jersey, United States of America,
**2** Princeton Neuroscience Institute, Princeton University, Princeton, New Jersey, United States of America

* cavargas@princeton.edu

**Data Availability Statement:** Supplementary material of this article, including code and data, is

## Abstract

A fundamental feature of the human brain is its capacity to learn novel motor skills. This capacity requires the formation of vastly different visuomotor mappings. Using a grid navigation task, we investigated whether training variability would enhance the flexible use of a visuomotor mapping (key-to-direction rule), leading to better generalization performance. Experiments 1 and 2 show that participants trained to move between multiple start-target pairs exhibited greater generalization to both distal and proximal targets compared to participants trained to move between a single pair. This finding suggests that limited variability can impair decisions even in simple tasks without planning. In addition, during the training phase, participants exposed to higher variability were more inclined to choose options that, counterintuitively, moved the cursor away from the target while minimizing its actual distance under the constrained mapping, suggesting a greater engagement in model-based computations. In Experiments 3 and 4, we showed that the limited generalization performance in participants trained with a single pair can be enhanced by a short period of variability introduced early in learning or by incorporating stochasticity into the visuomotor mapping. Our computational modeling analyses revealed that a hybrid model between model-free and model-based computations with different mixing weights for the training and generalization phases, best described participants' data. Importantly, the differences in the model-based weights between our experimental groups, paralleled the behavioral findings during training and generalization. Taken together, our results suggest that training variability enables the flexible use of the visuomotor mapping, potentially by preventing the consolidation of habits due to the continuous demand to change responses.

## Author summary

The development of new motor skills often requires the learning of novel associations between actions and outcomes. These novel mappings can be flexible and generalize to new situations, or more local with narrow generalization, similar to stimulus-action associations. In a series of experiments using a navigation task, we showed that generalizable

available as a project page on the Open Science Framework at https://osf.io/trjsg/.

**Funding:** The research reported in this manuscript was supported by the National Institute of Neurological Disorders and Stroke of the National Institutes of Health R01NS131552 (awarded to JT). This work was also supported by the Office of Naval Research N00014-18-2873 (awarded to JT), J. Insley Blair Pyne Fund (awarded to JT), Cognitive Science Program (awarded to CV), and Research Innovation Fund for New Ideas in the Natural Sciences (awarded to JT) at Princeton University. The funders had no role in study design, data collection and analysis, decision to publish, or preparation of the manuscript.

**Competing interests:** The authors have declared that no competing interests exist.

mappings are favored under a training variability regime, while local mappings with narrow generalization are developed in the absence of variability. Training variability was generated in our experiments either with multiple goals or with stochasticity in the action-outcome mapping, with both regimes leading to successful generalization. In addition, we showed that the benefits in generalization from training variability can be observed even when participants are subsequently exposed to no variability for a prolonged period of time. These results were best described by a mixture of model-free and model-based reinforcement learning algorithms, with different mixture weights for the training and generalization phases.

## Introduction

The first problem to be overcome in learning any novel motor skill is to associate particular actions with desired outcomes. This problem has become increasingly complex in the digital age, where the mapping between actions and outcomes can be as diverse as the imagination allows–just consider the variety of action-outcome associations underlying digital applications and video games. For example, using two thumbs to type a text message, using a pinch motion to zoom in and out of content on a smartphone, or steering a car in a video game. At first, learning these novel mappings is cumbersome and effortful but as learning progresses a mapping between actions and outcomes is eventually formed, allowing the individual to use the device successfully with ease. The formation of this mapping is arguably one of the most important steps for learning any new motor skill [1–5]. Surprisingly, however, we know very little, with a few exceptions [6,7], about how novel motor mappings are initially formed.

Traditionally, the question of how motor mappings are learned has been the focus of sensorimotor adaptation tasks (e.g., prisms, visuomotor rotations, and force fields), which impose a perturbation on the sensory outcome of a movement [8–10]. While adaptation tasks were originally thought to serve as a model paradigm to study this question [11–13], in recent years, it has become clear that these tasks may only pressure the recalibration of an existing motor mapping when faced with an externally imposed perturbation–not the establishment of the mapping in the first place [14,15]. Only when these recalibration mechanisms fail to fully counteract these perturbations in adaptation paradigms [15], may more *de novo* learning engage to develop a new controller for the task [16].

While there have been numerous studies of operant conditioning and associative learning, linking actions to outcomes, it is unclear the degree to which learning in these studies reflects the formation of a motor mapping *per se* [17–19]. It can be helpful, at least conceptually, to distinguish two levels of choice: a more abstract, internal level of reasoning about goals and state changes, and a more external, response-focused level about how to use movements to bring these plans to fruition. In many studies, such as in spatial or maze navigation, the agent already knows the control policy of how to move (i.e., how an action leads to a state change) and instead the focus is on reasoning or learning at a more abstract level how the state change leads to a desired outcome in terms of reward [20,21]. Conversely a different set of paradigms focus entirely on externally-cued responses, without any internal plan. Such tasks include motor sequence learning [22–24], discrete sequence production [25,26], and *m x n* tasks [27,28], all of which can be viewed as a form of *de novo* motor learning, establishing a relationship between arbitrary actions and outcomes. However in these studies, there is no underlying mapping from internal goals, from which a generalizable, motor map may form.

Generalization to new situations or contexts is considered a hallmark feature of a motor mapping, as opposed to rote memorization of stimulus-response associations [8,29,30].

Furthermore, the different variants of these sequence learning tasks are externally generated, such that the appropriate sequence of responses is fully specified by the experimental stimuli [31]. The participant must precisely follow the set of stimulus-response pairs to be successful in the task. As such, they may only reflect a subset of the kinds of motor skills that we perform in everyday life that are internally generated. Thus, while there has been tremendous progress in understanding how externally-generated, stimulus-response mappings are learned, there has been comparatively less progress in understanding how internally-generated, response-outcome mappings are formed.

The increased complexity and degrees of freedom available when learning an internally-generated, response-mapping may be one potential reason why progress has been slow. The difficulty is in designing and studying a task that is in the "Goldilocks zone" between sufficient experimental complexity and analytical tractability [32]. Fermin and colleagues developed the *grid navigation task* to fit within this zone to study the core problem of learning an internally-generated mapping: The formation of a novel and arbitrary motor mapping [33,34]. Here, participants learn to navigate a cursor from various starting locations to various target locations on a grid through a series of keypresses. The goal is to navigate to the target location in the minimum number of moves as quickly as possible. Importantly, there is an unintuitive and arbitrary mapping between the keys and cursor movement that must be learned.

While in simple versions of this task participants can learn within a relatively short period, it remains an open question as to how this is accomplished [31,33,35]. The formation of a new mapping is not always guaranteed. If the task only demands the repetition of a limited set of actions, then only local state-action associations may be learned–a form of rote memorization, which is likely what occurs in studies of sequence learning. However, if there is a greater degree of variability in training, then a richer representation of skill may be learned, such as the formation of an internal model between the action-outcome space. This would afford the ability to generalize outside the range of training [36–41]–an idea that echoes classic theories of stimulus variability in learning [42,43]. These two forms of learning mirror the instance-based and algorithmic processes of a classic theory for automatization [44], as well as the more modern notions of model-free and model-based reinforcement learning [20,45,46,47]. In particular, the latter formalism seems well suited to capturing the candidate mechanisms. Model-based reinforcement learning is well suited to capture the covert formation of an abstract, internal plan that can then be generalizably realized through a separately learned motor mapping. This leads to the hypothesis that, much as in other circumstances such as operant leverpressing [45], simpler model-free (stimulus-response) learning will instead dominate when a narrow range of actions is overtrained.

Here, through a series of experiments, we seek to build on this work by leveraging the grid navigation task as a model paradigm to study how novel and arbitrary motor mappings are initially formed and seek to characterize how they may be learned through the model-free and model-based reinforcement learning framework. We hypothesize that the formation and representation of a novel motor mapping depends on the particular conditions of training. Specifically, the degree of exploration between the number of potential action goals and possible solutions to achieve that goal may pressure formation of a generalizable motor mapping over local state-action associations (e.g., rote memorization of specific sequences of actions). Generalization to untrained conditions will provide a key test for the existence of a motor mapping.

## Results

### Experiment 1

How does training variability constrain learning and generalization of a visuomotor mapping? Two groups of participants performed a grid navigation task [33,34] (Fig 1A) where they moved a cursor from various start to target locations using the J, K and L keys of a standard keyboard (see *Materials and Methods* for details). In the Single group (n = 16), participants trained to move between a single start-target pair, while in the Multiple group (n = 16) participants trained to move between four start-target pairs (Fig 1B). We predicted that performance improvements for the Multiple group would be slower during training, but they would be able to generalize their performance to novel start-target pairs, reflecting the formation of a key-to-direction mapping rather than local state-action associations. In contrast, the Single group would show faster performance improvements during training, but be unable to generalize to new start-target pairs. Participants performed 260 training trials followed by 20 trials of generalization interleaved with 20 training trials.

### Behavioral results

Fig 2A shows the proportion of optimal arrivals over trial bins (1 bin = 10 trials) for both groups, where an optimal arrival is a trial where subjects arrive at the target using the

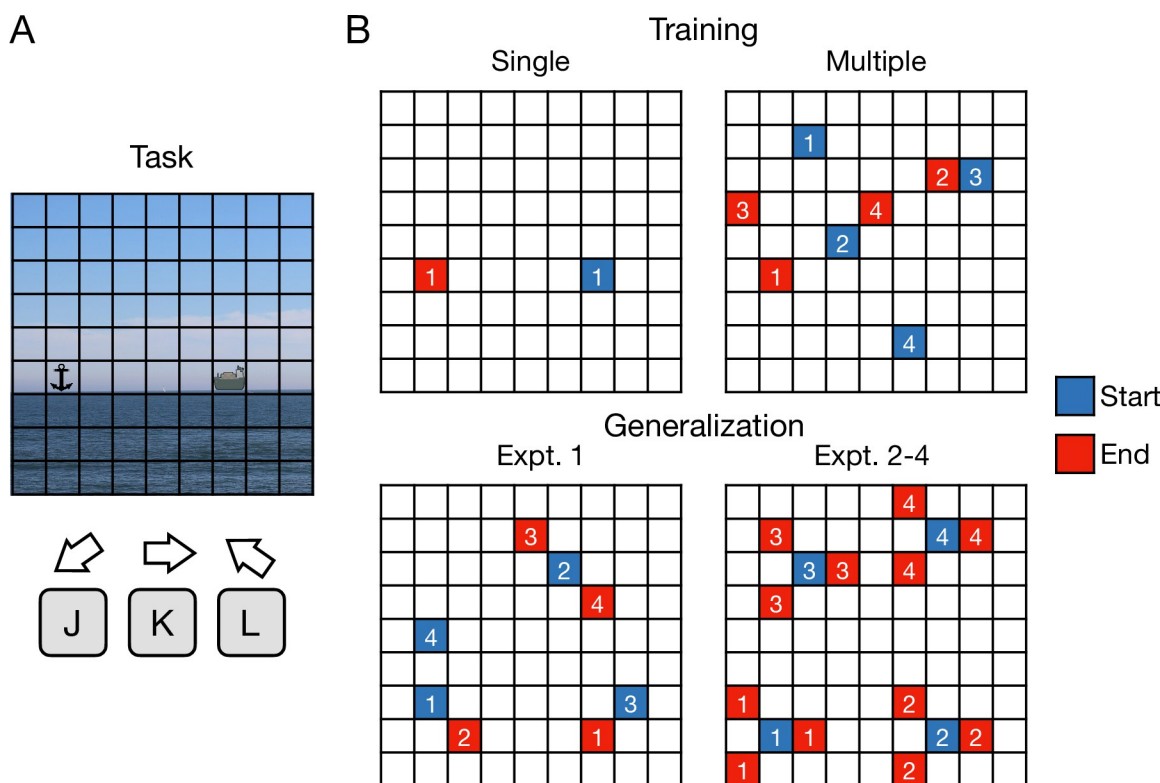

**Fig 1. Experimental task. (A)** Participants moved a cursor (ship) from start to target (anchor) locations in a grid environment. For Experiments 1–3, participants used a deterministic visuomotor mapping of three keys with moving directions: bottom-left, right and top-left. In Experiment 4, the mapping randomly changed after each keypress with a probability of 0.2 to any of the remaining directions. **(B)** In the four experiments, participants were trained with a single or multiple start-target pairs (see *Materials and Methods* for details). A generalization phase was presented after training where the target locations were either seven (Experiment 1) or one move away (Experiments 2–4) from the starting point. Blue and red grid states represent the potential start and target locations where the numbers indicate a specific pair. Only one pair was presented per trial. Images for the task were obtained from the open source website *https://commons.wikimedia.org*.

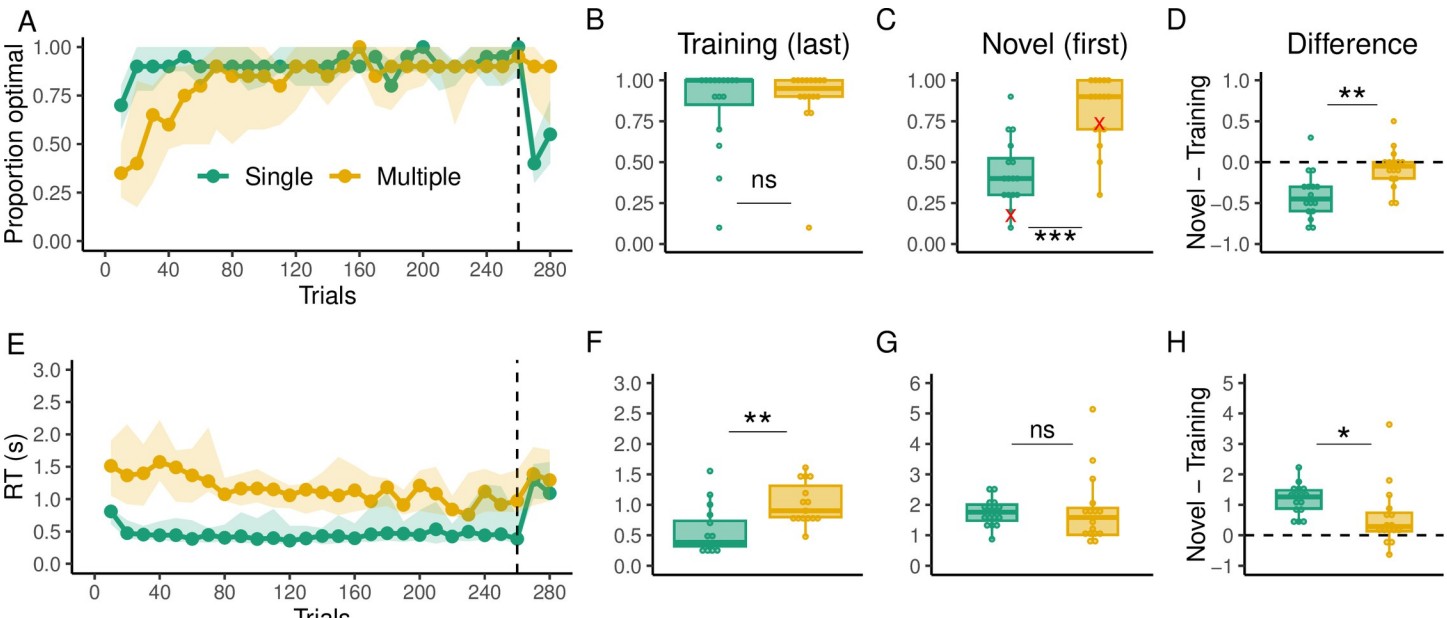

**Fig 2. Behavioral results of Experiment 1.** (A) Proportion of optimal arrivals over trial bins for the Single (green) and Multiple (gold) groups. The black dashed line indicates the beginning of the generalization phase. The solid dotted line represents the median and the shading region the interquartile range. (B) Proportion of optimal arrivals in the last bin of the training phase. (C) Proportion of optimal arrivals in the first bin of generalization (novel pairs). Red marks indicate performance in the very first trial of generalization for all subjects. (D) Difference in the proportion of optimal arrivals between the first bin of generalization and the last one of training. The dashed line here indicates no performance change from training to generalization (E) RTs over trial bins. (F) RTs in the last bin of training trials. (G) RTs in the first bin of generalization (H) Difference in RTs between the first bin of generalization and the last one of training.

minimum number of key presses (7 moves). As expected, the Multiple group had a slower learning curve as revealed by a mixed-effects model analysis. Specifically, the Single group demonstrated significantly higher performance compared to the Multiple group up to the bin corresponding to trial 60 ($p < 0.05$). However, both the Single and Multiple groups reached the same level of performance by the end of the training phase (comparison of the last bin of training trials between groups; $t(29.06) = -0.5$, $p = 0.61$; Fig 2B). In addition, in the Multiple group, there were no substantial differences in performance among the different start-target pairs, with optimal arrivals generally increasing over time (S1 Fig). Of more importance is how well the groups perform when new start-target pairs are introduced in the generalization phase. This was determined by quantifying any potential change in performance between the end of the training phase and beginning of the generalization phase. For the Single group, we found that performance was significantly worse at the onset of the generalization phase compared to the end of the training phase ($t(15) = -5.67$, $p < 0.001$; Fig 2D). In contrast, the Multiple group's performance during generalization was similar to the end of the training phase ($t(15) = -1.21$, $p = 0.242$). When comparing between the groups, it was clear that the Multiple group performed significantly better than the Single group early in generalization (first bin; $t(29.99) = -5.1$, $p < 0.001$; Fig 2C) and still marginally better at the end of the generalization (second bin; $t(29.27) = -1.64$, $p = 0.05$). The change in performance between the groups was significantly different ($t(29.34) = -3.5$, $p = 0.001$; Fig 2D).

We also examined how planning (reaction time, RT) evolved over training. RTs were overall higher in the Multiple group group throughout training ($t(29.99) = -4.82$, $p < 0.001$; Fig 2E) and higher at the last bin of training trials ($t(21.21) = -2.85$, $p = 0.009$; Fig 2F) but not during the first bin of generalization trials ($t(19.36) = -0.17$, $p = 0.86$; Fig 2G). There was an increase in RTs from the last training bin to the first generalization bin in the Single ($t(15) = 9.54$,

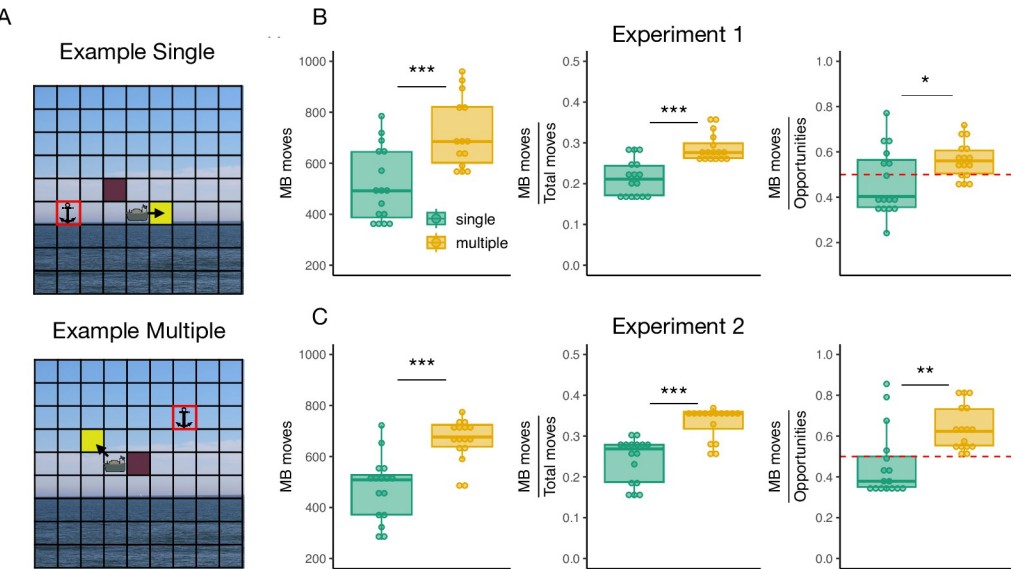

**Fig 3. Behavioral evidence of model-based computations during training of Experiment 1 and Experiment 2.** (A) Examples in the Single (top) and Multiple (bottom) groups where the cursor can move to two states that minimize the distance to the target. While moving to the purple state minimizes both the chessboard distance (getting visually closer to the target) and the model-based distance (getting closer to the target in mapping space), moving to the yellow state (MB move) minimizes only the true, model-based distance, i.e., the cursor moves visually away from the target but closer in mapping space. (B) Left: Total number of MB moves during training for each group. Middle: Number of MB moves relative the total number of moves in the experiment. Right: Proportion of MB moves relative to the number of opportunities to make that choice, i.e., choosing to minimize the model-based distance over the chessboard distance as in (A). (C) The same as in (B) but for Experiment 2. The red dashed lines indicate chance level when choosing to minimize the chessboard distance or the model-based distance. Images for the task were obtained from the open source website *https://commons.wikimedia.org*.

p < 0.001; Fig 2H) and Multiple groups (t(15) = 2.34, p = 0.03), however, this increase was significantly greater in the Single group (t(21.96) = 2.14, p = 0.043). In addition, the groups only differ in RTs over training, but not in any of the inter-key-intervals (i.e., the time in between key presses; S2 Fig). Crucially, however, we found that during the generalization phase, the Single group had significantly higher inter-key-intervals than the Multiple group (S3 Fig), suggesting that they had to replan at intermediate steps of the trajectory unlike the Multiple group. For the Multiple group, we did not find substantial differences in terms of RTs among the different start-target pairs in terms of RTs. In general, RTs decreased over time in all pairs (S1 Fig).

Differences in generalization performance provide preliminary evidence that the Multiple group could have used a more flexible representation of the mapping compared to the Single group to arrive at the novel targets. In the following analysis we asked whether a similar distinction could have arisen during the training phase. To investigate this, we hypothesized that participants who rely on the visuomotor mapping would make more frequent choices that require planning, compared to participants relying primarily on state-action values. In particular, the former would be more inclined to choose options where the cursor counterintuitively moves away from the target while actually reducing the true, model-based distance (Fig 3A). While choices that move the cursor visually closer to the target can result from heuristics that minimize unconstrained distance metrics (e.g., the chessboard distance), choices that minimize only the model-based distance ("MB move") require considering the long-term outcomes of using the constrained mapping.

Indeed, we found that participants in the Multiple group made more MB moves in the training phase compared to the Single group (t(29) = -4.34, p < 0.001; Fig 3A, left), and also relative to the total number of moves (t(28.17) = -5.26, p < 0.001; Fig 3A middle), which controls that participants in the Multiple group made on average more key presses in the experiment. Most notably, we showed that this pattern remained when considering the opportunities that participants had to minimize the model-based distance over the chessboard distance (t(23.46) = -2.34, p = 0.02; Fig 3A, right). The latter analysis excludes situations where minimizing the chessboard distance or the model-based distance are confounded. For example, when the target is only one move away from the cursor, the winning move minimizes both distances while the other two moves increase them; similarly, when the target is to the right of the cursor and on the same row, pressing the key that moves the cursor to the right is the only move that minimizes both distances while the other two moves increase them. Such situations do not represent a choice opportunity in our analysis. When participants in the Multiple group had the opportunity to minimize either the model-based distance or the chessboard distance, they chose to minimize the former more often than chance (i.e, more than 50% of the times; t(15) = 3.09, p = 0.007), while in the Single group they did not (t(15) = -0.95, p = 0.35).

In addition, we also examined the most frequently used trajectories across participants during the training phase (S4 Fig). We found that the preferred trajectories were less likely to be shared by other participants in the Single group compared to the Multiple group. In particular, between 3–5 participants (18–31%) in the Single group shared the same preferred trajectory. In contrast, between 7–11 participants (43–68%) in the Multiple group converged to the same solution for the majority of the trials. The greater heterogeneity in the trajectories used by the Single group likely resulted from the requirement to utilize all three keys to arrive at the target. On the other hand, the Multiple group only required the use of two keys, although all keys were needed across the experiment. In fact, using a dynamic programming approach, we determined that there were a total of 140 trajectories that optimally reached the target in the Single group, whereas in the Multiple group, there were only between 21 and 35 optimal trajectories depending on the start-target pair. The larger number of successful trajectories in the Single group likely contributed to fewer participants finding the same solutions.

In a related analysis, we found that, within participants, both the Single and Multiple groups showed highly homogeneous trajectories during the training phase. In particular, participants in the Single group reliably chose two trajectories more than 50% of the time (t(15) = 2.86, p = 0.005; S5 Fig), whereas the Multiple group exclusively selected a single trajectory more than 50% of the time for each of the four start-target pairs (t(15) = [3.28, 2.56, 2.54, 2.05], p = [0.002, 0.01, 0.01, 0.02]). While both groups displayed highly repetitive behavior relative to their target, the absolute frequency of repetitions of the preferred trajectories was much smaller in the Multiple group given that, by design, each target appeared for fewer trials. Indeed, the most frequently used trajectory in the Single group was used significantly more times than the maximum number of times the trajectory in the Multiple group could have been used (i.e., 25% of the time; t(15) = 4.36, p < 0.001).

The differences in generalization performance between the Single and Multiple groups, which remained marginally significant even by the end of the generalization phase, suggest that the training regimes might have led to different learning representations in the task. While both groups chose highly repetitive trajectories during training, which aligns with the memorization of few solutions, the variability experienced by the Multiple group could have simultaneously allowed the better learning of the visuomotor mapping. Evidence supporting this comes from the more frequent selection of moves that exclusively reduce the model-based distance by the Multiple group. In addition, the greater repetition of the preferred trajectories in the Single group given its unique target, could have facilitated the formation of habitual

responses which interfered in the generalization phase. Indeed, supplementary analysis revealed that errors in the generalization phase of the Single group more often began with the same key as the one most frequently used trajectory during the training phase, compared to the Multiple group (S6 Fig). While these differences are numerical due to the limited power resulting from the low error rate in the Multiple group, they suggest that beside a greater representation of the mapping, training variability could have also prevented the interference from habitual responses.

## Modeling results

In order to explore the cognitive processes that gave rise to the differences in performance between the groups, we evaluated five computational models from the reinforcement learning literature (see *Materials and Methods* for details). On one end of the spectrum, we tested a model-free reinforcement learning algorithm [20] which learns state-action values using prediction errors based on the chessboard distance to the target. On the other end of the spectrum, we implemented a model-based algorithm that learns the visuomotor mapping of the task and uses it to find the shortest route to the target using tree search. We selected these models as they make contrasting predictions about generalization performance in the task (Fig 4). A fully trained model-free reinforcement learning algorithm would have good performance for familiar target locations, but generate chance-level responses for targets that have not been experienced in the past, thus predicting poor generalization. A model-based algorithm would similarly perform well for familiar targets but would instead be able to generalize to novel targets once the visuomotor mapping has been learned. Finally, we tested hybrid models between the model-free and model-based algorithms, which are weighted sums of the two components. Hybrid models would perform well in familiar targets as the model-free and model-based algorithms, but would be able to capture different degrees of generalization (Fig 4).

Based on previous work [45,46,48–50], we considered that the mixing weight between the model-free and model-based components in the hybrid model could take the following forms: a single weight across the entire experiment (1W), one weight for the training phase and one

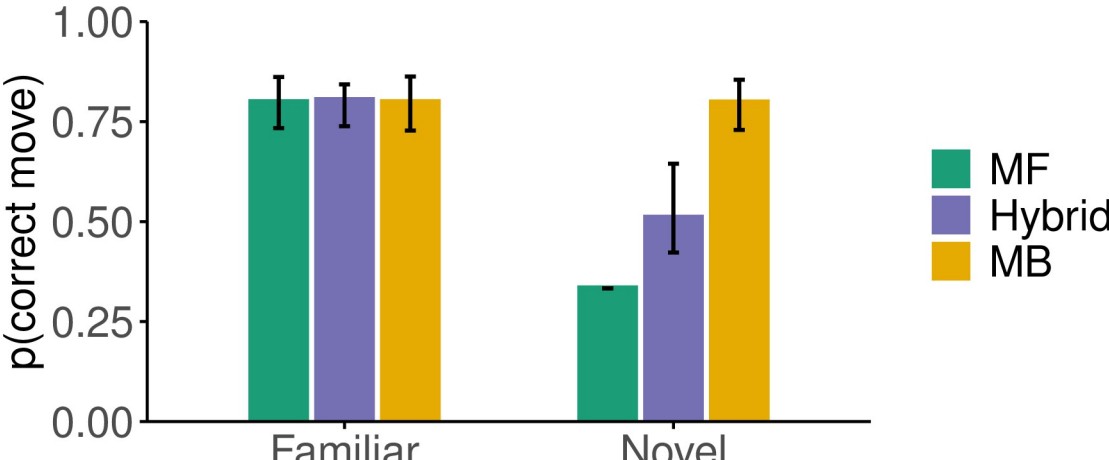

**Fig 4. Models simulation.** Probability of choosing the optimal move for the fully trained models on a familiar and novel target. All models are capable of achieving ceiling performance for familiar targets after sufficient training similar to human subjects. Responses were generated using an inverse temperature parameter $\beta = 4$ for all models. For the hybrid model 100 Uniform(0,1) samples of the model-based weight were used to simulate its performance. The height of the bars represents the median and the error bars the interquartile range.

**Table 1. Individual model comparison for Experiment 1.** The AIC and BIC columns show the median across subjects and the interquartile range inside the square brackets. AIC wins and BIC wins columns indicate the number of participants where the given model was the best one according to each metric.

| Groups | Models | AIC | AIC wins | BIC | BIC wins |
|--------|--------|-----|----------|-----|----------|
| Single | MF | 2637 [2037, 3563] | 0/16 | 2649 [2048, 3922] | 0/16 |
| | MB | 3968 [3339, 4555] | 0/16 | 3973 [3345, 4056] | 0/16 |
| | 1W | 2589 [1980, 3382] | 1/16 | 2612 [2003, 3665] | 1/16 |
| | 2W | 2556 [1968, 3359] | 14/16 | 2584 [1996, 3668] | 10/16 |
| | AR | 2585 [1965, 3381] | 1/16 | 2608 [1988, 3677] | 5/16 |
| Multiple | MF | 3079 [2508, 3910] | 0/16 | 3090 [2519,3922] | 0/16 |
| | MB | 3109 [2631, 4050] | 0/16 | 3115 [2637, 4056] | 0/16 |
| | 1W | 2766 [2272, 3641] | 1/16 | 2790 [2294, 3665] | 5/16 |
| | 2W | 2753 [2263, 3639] | 11/16 | 2782 [2291, 3668] | 4/16 |
| | AR | 2761 [2273, 3653] | 4/16 | 2784 [2296, 3677] | 7/16 |

for the generalization phase (2W), and a time varying weight. For the latter model, we developed a Bayesian arbitration mechanism based on the history of familiar or novel states (AR model; see *Materials and Methods* for details) which assigns a greater weight to the model-based component if novel states are frequently encountered. On the other hand, if familiar scenarios are experienced, the model-based weight decreases, giving way to the more habitual, model-free system. This transition from model-based to model-free control has been reported previously [48], and a preliminary modeling analysis revealed that this was a plausible scenario in our experiment given the task statistics of the Single and Multiple groups (S7 and S8 Figs). In order to compare the models, we used the Akaike Information Criterion (AIC) [51] and the Bayesian Information Criterion (BIC) [52]. The results of the model comparison are shown in Table 1.

For the Single group, the 2W hybrid model best described the majority of participants according to both the AIC (14 out of 16) and the BIC (10 out of 16) metrics. In contrast, the Multiple group showed mixed results depending on the metric used. While the 2W hybrid model best described the majority of participants according to the AIC (11 out of 16), the BIC indicated that the arbitration hybrid model was the best for 7 out of 16 participants, followed by the 1W hybrid model (5 out of 16) and the 2W hybrid model (4 out of 16). To provide a global metric of model performance across subjects, we performed a model comparison at the group level [53] (see *Materials and Methods* for details; Fig 5A). This analysis indicated that the 2W hybrid model provided the best overall description for both groups. Specifically, a subject taken at random from the Single group had a 98% probability of being best described by the 2W hybrid model, followed by 2% from the arbitration model. For the Multiple group, this probability was 57% for the 2W hybrid model, followed by 31% for the arbitration hybrid model, and 12% for the 1W hybrid model.

In addition, we computed the probability that a given model was more likely than the others in the population, i.e. the exceedance probability [53]. According to this metric, there was a high probability (>99%) that the 2W hybrid model was better than the rest of the models in both groups. In addition, in order to evaluate how good the models were in the absolute sense, we computed the proportion of the variability in our data that was explained by our models as compared to the negative entropy, which is a near upper bound for our probabilistic models [54–56] (see *Materials and Methods* for details). The median proportion of the variability explained by the best model according to the BIC in the Single group was 76%, whereas in the Multiple group it was 68% (Fig 5B).

Given that the 2W hybrid model provided the best description of the data in the aggregate, we show the weights toward the model-based component during training and generalization

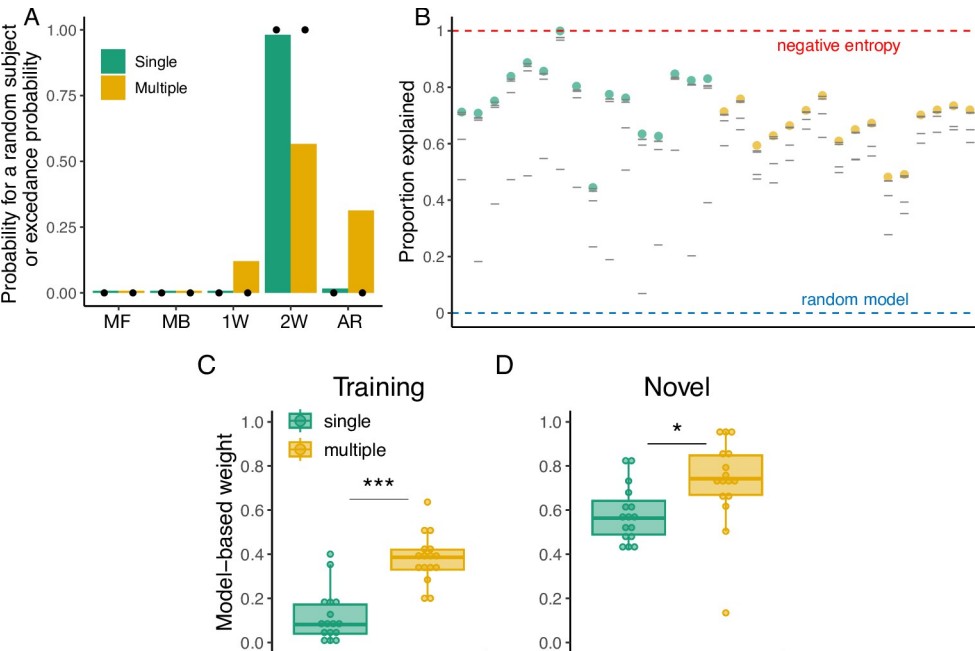

**Fig 5. Modeling results of Experiment 1. (A):** Probability that a random subject taken from the Single and Multiple groups is best described by the tested models (MF: model-free, MB: model-based, 1W: Single-seight hybrid, 2W: Two-weight hybrid, AR: arbitration model). Black dots indicate the exceedance probability. **(B)** Proportion of the variability in the data explained by the models. Colored dots represent this value for the best model according to the BIC and gray lines represent the other models. Red and blue dashed lines represent the negative entropy (upper boundary) and the performance of a random model (lower boundary), respectively. **(C)** Model-based weight in the hybrid model during training and **(D)** generalization.

for both the Single and Multiple groups (Fig 5C and 5D). Differences in the weights can indicate whether participants relied more on the model-free system during training due to repetitions but switched to the model-based system when the task demands changed. We performed a 2-Way ANOVA with the experimental phase (training or generalization) and the group (Single or Multiple) as factors, and found a significant main effect of the experimental phase ($F(1,60) = 124.56$, $p < 0.001$, $\eta^2 = 0.56$), indicating that overall participants during the generalization phase had higher weights for the model-based component compared to the training phase. Similarly, we found a significant main effect of the group ($F(1,60) = 31.9$, $p < 0.001$, $\eta^2 = 0.43$), where the Multiple group had higher model-based weights overall. A *post hoc* pairwise analysis using the Tukey's Honest Significant Difference (HSD) found that the Multiple group had significantly higher model-based weights during training (adjusted $p < 0.001$) and generalization (adjusted $p = 0.02$) compared to the Single group. These results corroborate our behavioral findings, where participants in the Multiple group tended to reduce the model-based distance to the target more often than the Single group during training (Fig 3B), and also that they had better generalization performance (Fig 2D).

Overall, the behavioral and modeling results from Experiment 1 suggest that greater variability during training enhances the use of the visuomotor mapping in novel scenarios, and potentially reduces the interference from habitual responses. These behavioral findings are supported by our computational modeling results, where we found greater model-based weights in the Multiple group. Nevertheless, it is relevant to note that other factors could have contributed to such results. In particular, the Multiple group could have shown better generalization performance if the trajectories used during training were more similar to the ones

required to arrive at the targets during generalization as compared to the Single group. Second, while the start-target pairs in generalization trials were novel, providing feedback about the cursor movement in such trials could generate learning about the mapping, which creates a confound between new learning and generalization. Lastly, it is unclear whether differences in generalization performance between the groups occurred because of a distinct representation at the level of the visuomotor mapping or because the groups differed in their ability to use the mapping sequentially (planning), with the Single group being unable to generate appropriate sequences with an otherwise known mapping. We address these concerns in Experiment 2.

## Experiment 2

In Experiment 1, we found that the Multiple group readily generalized their performance to new start-target pairs during the generalization phase, while the Single group struggled to generalize to novel pairs. However, feedback was available during the generalization phase, allowing the Single group to potentially relearn the mapping to recover performance, which is suggested by the elevated RTs during this phase. Here, we sought to control for this possibility by not providing feedback during the generalization phase and placing the target only one move away from the starting location (Fig 1B). In addition, to prevent further learning during the generalization phase, we also removed the interleaved training trials.

Removing the sequential component from the generalization phase can also rule out the potential confound that differences in performance arose due to distinct capabilities to plan new movement sequences resulting from the training regime. Finally, placing the target one step away, removes the possibility that better generalization performance occurs due to greater similarity between the sequences generated during training and the ones required in generalization.

If participants in the Single group still underperform the Multiple group in this simple situation, it would provide further evidence that they did not know or could not use the mapping to the extent that the Multiple group could by the end of the training phase. Apart from these changes, everything else remained as in Experiment 1. There were a total of 260 trials of training for both the Single (n = 16) and Multiple (n = 16) groups and 20 trials of generalization.

## Behavioral results

Similar to Experiment 1, we found that the Multiple group had a slower learning curve (Fig 6A) as revealed by a mixed-effects model where the Single group had significantly higher performance up until the bin corresponding to trial 50 (p < 0.05). However, both groups reached the same level of performance by the end of the training phase (t(29.99) = -0.19, p = 0.84; Fig 6B). In addition, no apparent difference in performance was noticed among the different start-target pairs of the Multiple group, with optimal arrivals generally improving with time (S9 Fig). Of primary interest is how each group performed during the generalization phase where only one move was required and no feedback was provided. Notably, the Multiple group still outperformed the Single group early (first bin; t(22.68) = -4.81, p < 0.001; Fig 6C) but also late in the generalization phase (second bin; t(19.41) = -3.1, p = 0.005). However, the performance of the Single group remained greater than chance (t(15) = 5.88, p < 0.001), which suggests that even though they performed worse than the Multiple group, they could have recalled some knowledge about the mapping. In addition, the Single group's performance was significantly worse at the onset of generalization (first bin; t(15) = -5.03, p < 0.001; Fig 6D) compared to late in training, but the Multiple group's performance did not significantly decrease between the training and generalization phases (t(15) = -1.24, p = 0.23). The continued worse performance by the Single group in the generalization phase suggests that they struggled to effectively

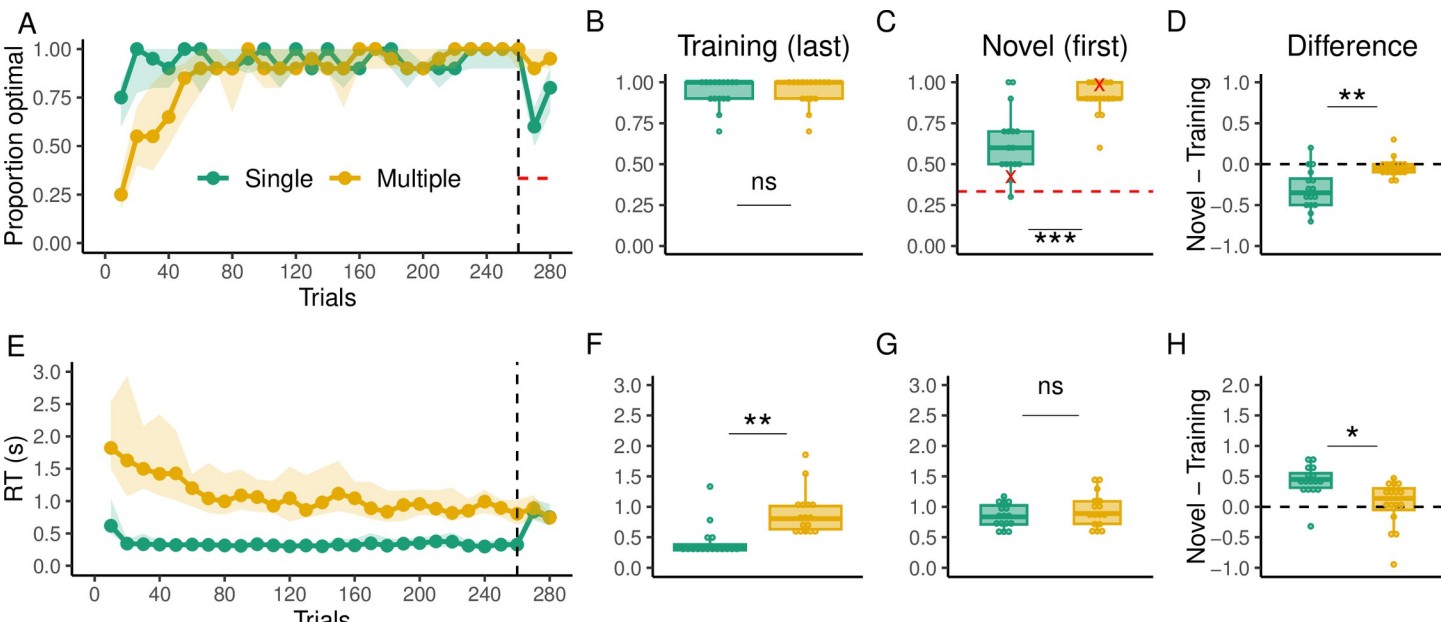

**Fig 6. Behavioral results of Experiment 2.** (**A**) Proportion of optimal arrivals over trial bins for the Single (green) and Multiple (gold) groups. The black dashed line indicates the beginning of the generalization phase. The solid dotted line represents the median and the shading region the interquartile range. The red dashed line indicates chance level of performance. (**B**) Proportion of optimal arrivals in the last bin of training trials. (**C**) Proportion of optimal arrivals in the first bin of generalization (novel pairs). Red marks indicate performance in the very first trial of generalization for all subjects. (**D**) Difference in the proportion of optimal arrivals between the first bin of generalization and the last one of training. The dashed line here indicates no performance change from training to generalization (**E**) RTs over trial bins. (**F**) RTs in the last bin of training trials. (**G**) RTs in the first bin of generalization (**H**) Difference in RTs between the first bin of generalization and the last one of training.

switch to model-based control and that variability in training may be necessary for a model-based algorithm to be efficiently used.

Similar to Experiment 1, RTs in the Multiple group were overall higher during training (Fig 6E; t(16.06) = -8.31, p < 0.001) and higher at the last bin of training trials (Fig 6F; t(28.13) = -4.22, p < 0.001) but not in the first bin of generalization trials (Fig 6G; t(26.48) = -1.08, p = 0.28). RTs significantly increased from the last training bin to the first generalization bin in the Single (t(15) = 6.55, p < 0.001) but not in the Multiple group (t(15) = 0.41, p = 0.68). In addition, this change in RTs was significantly greater in the Single group (t(26.43) = 3.34, p = 0.002; Fig 6H).The differences in RTs increase suggest that the Multiple group did not experience a switch in the algorithm used to solve the task, whereas the Single group likely transitioned to a more computationally demanding algorithm. As in Experiment 1, we did not find noticeable differences in RTs among the different start-target pairs of the Multiple group, with RTs generally decreasing over time (S9 Fig).

Subsequently, we performed the same analysis as in Experiment 1 looking for behavioral signals of the use of the visuomotor mapping over state-action associations during training. Corroborating our previous results, we found that participants in the Multiple group generated more moves that minimized exclusively the model-based distance to the target in the absolute sense (t(26.29) = -5.02, p < 0.001; Fig 3C, left), relative to the total number of moves in the experiment (t(27.71) = -5.45, p < 0.001; Fig 3C, middle) and relative to the opportunities they had to make this choice (t(25.66) = -3.55, p = 0.001; Fig 3C, right). Likewise, we found that in situations where participants could minimize either the model-based distance or the chess-board distance, they chose to minimize the former significantly more than chance (50% of the times) in the Multiple group (t(15) = 5.12, p < 0.001; Fig 3C, right) but not the Single group (t (15) = -0.92, p = 0.37).

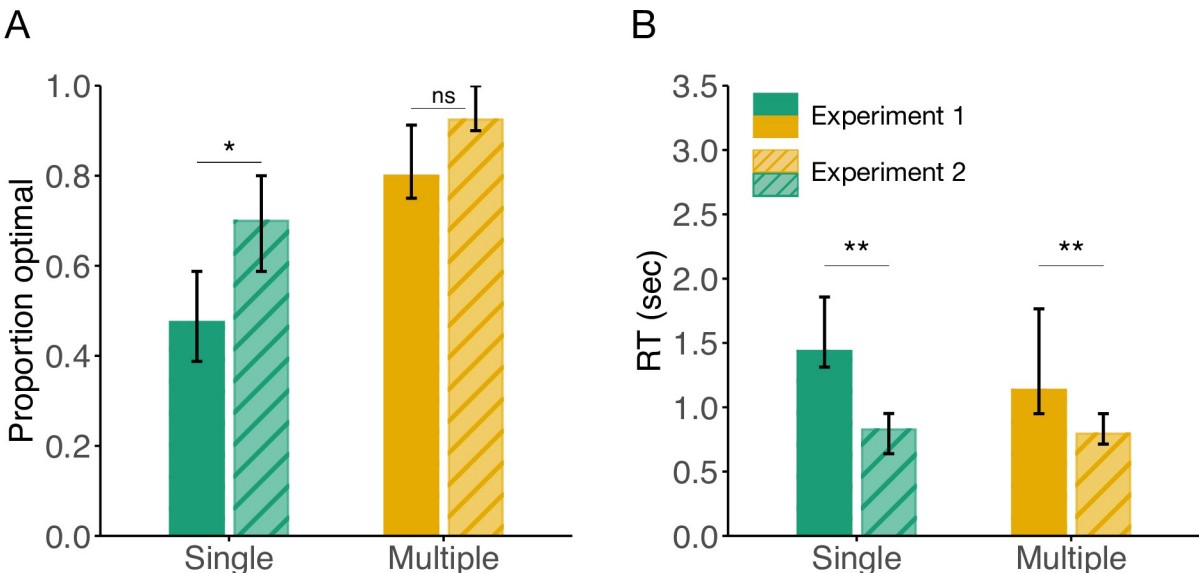

**Fig 7. Comparison of generalization performance between Experiment 1 (smooth bars; target seven moves away) and Experiment 2 (striped bars; target one step away) for the Single and Multiple groups. (A)** Proportion of optimal arrivals and **(B)** reaction times for all trials in the generalization phase. The height of the bars represents the median and the error bars the interquartile range.

In subsequent analyses, aiming to compare the generalization performance of participants in Experiment 1 (targets seven moves away) and Experiment 2 (targets one move away), we found converging evidence that participants in the Multiple group performed overall better. Specifically, we carried out a 2-Way ANOVA over the proportion of optimal arrivals across all generalization trials, with experiment number (Experiment 1 and Experiment 2) and group (Multiple and Single) as factors, which revealed a significant main effect of group ($F_{(1,60)}$ = 31.07, p < 0.001, $\eta^2$ = 0.29; Fig 7A). Similarly, we found a significant main effect of experiment number ($F_{(1,60)}$ = 13.28, p < 0.001, $\eta^2$ = 0.12), where participants in Experiment 2 performed overall better. *Post hoc* pairwise comparisons revealed that while there were significant differences in generalization between Experiment 1 and Experiment 2 in the Single group (adjusted p <0.001), this difference did not reach significant levels for the Multiple group (adjusted p = 0.09), suggesting that planning primarily affected the optimal arrivals in the Single group.

We performed the same analysis overt the RTs of Experiment 1 and Experiment 2, and found a significant main effect of experiment number ($F_{(1,60)}$ = 27.61, p < 0.001, $\eta^2$ = 0.31; Fig 7B) but not of group ($F_{(1,60)}$ = 0.15, p = 0.69, $\eta^2$ = 0.04). The *post hoc* pairwise comparisons revealed that participants in both the Single (adjusted p = 0.004) and Multiple (adjusted p = 0.001) groups had lower RTs in the generalization trials of Experiment 2, indicative of a lower computational demand compared to Experiment 1 where more planning was necessary.

Regarding the trajectories that participants used during the training phase, we found similar results as in Experiment 1. In particular, participants in the Single group converged to different trajectories with only 2–3 participants (12–18%) sharing their most frequently used trajectory. In contrast, 7–8 participants (44–50%) in the Multiple group shared the most frequent trajectory (S4 Fig). As in Experiment 1, we observed repetitive behavior in both the Single and Multiple groups. Particularly, participants in the Single group chose a single trajectory more than 50% of the time ($t_{(15)}$ = 2.06, p = 0.02; S5 Fig). Similarly, the Multiple group selected a single trajectory more than 50% of the time ($t_{(15)}$ = [6.21, 6.23, 5.63, 3.91], p < 0.001). However, as in Experiment 1, the absolute frequency of repetitions for preferred trajectories was much

smaller in the Multiple group given that each target appeared for fewer trials. The most frequently used trajectory in the Single group was used significantly more times than the maximum number of times the trajectory in the Multiple group could have been used (i.e., 25% of the time; t(15) = 5.88, p < 0.001).

## Modeling results

As in Experiment 1, we evaluated five models: the model-free, model-based and three hybrid models. The results of the model comparison are presented in Table 2. According to the AIC, we found that the majority of participants were best described by the 2W hybrid model in the Single (9 out of 16) and Multiple groups (14 out of 16). However, some participants were best described by the rest of the models as well. In contrast, according to the BIC metric, there was less evidence for a dominant model at the individual level. In the Single group, the arbitration model best described the largest number of participants (6 out of 16), followed by the 2W hybrid model (5 out of 16), the model-free model (3 out of 16), and the 1W hybrid model (2 out of 16). For the Multiple group, the 2W hybrid model best described half of the participants (8 out of 16), followed by the 1W hybrid model (5 out of 16) and the arbitration model (3 out of 16).

When we considered the aggregate evidence for each model across participants, we found that a random subject taken from the Single group would have a 66% probability of being best described by the 2W hybrid, 27% by the arbitration model, 5% by the 1W hybrid model and 2% by the model-free model. For the Multiple group, there was a 92% probability that a random subject would be best described by the 2W hybrid model, followed by 6% from the 1W hybrid model and 2% from the arbitration model. According to the exceedance probability, there was a high probability (>99%) that the 2W hybrid model was better than the rest of the models in both the Single and Multiple groups. Thus, at the population level, the 2W hybrid was the predominant model overall (Fig 8A). For the best performing models at the individual level according to the BIC, we found that, when compared with the theoretical near upper bound, they captured 73% and 68% of the explainable variability in the data in the Single group and Multiple group, respectively (Fig 8B).

Similar to Experiment 1, we analyzed the weights of the 2W hybrid model during training and generalization. A 2-Way ANOVA with experimental phase (training or generalization) and group (Single or Multiple), showed a significant main effect of the experimental phase (F (1,60) = 111.8, p < 0.001, $\eta^2$ = 0.48) and the group (F(1,60) = 60, p < 0.001, $\eta^2$ = 0.26), indicating that overall participants had higher model-based weights during generalization and in the Multiple group. Post hoc pairwise comparisons revealed that the model-based weight was

**Table 2. Individual model comparison for Experiment 2.** The AIC and BIC columns show the median across subjects and the interquartile range inside the square brackets. AIC wins and BIC wins columns indicate the number of participants where the given model was the best one according to each metric.

| Groups | Models | AIC | AIC wins | BIC | BIC wins |
|---|---|---|---|---|---|
| Single | MF | 2090 [1526, 2603] | 1/16 | 2101 [1537, 2614] | 3/16 |
| | MB | 3204 [2775, 3526] | 0/16 | 3209 [2780, 3532] | 0/16 |
| | 1W | 2074 [1498, 2482] | 1/16 | 2096 [1520, 2505] | 2/16 |
| | 2W | 2069 [1470, 2465] | 9/16 | 2097 [1497, 2493] | 5/16 |
| | AR | 2072 [1484, 2479] | 5/16 | 2094 [1507, 2502] | 6/16 |
| Multiple | MF | 2468 [2063, 2854] | 0/16 | 2497 [2074, 2865] | 0/16 |
| | MB | 2436 [2164, 2780] | 0/16 | 2441 [2169, 2785] | 0/16 |
| | 1W | 2157 [1845, 2551] | 1/16 | 2197 [1868, 2573] | 3/16 |
| | 2W | 2132 [1838, 2543] | 14/16 | 2160 [1865, 2571] | 8/16 |
| | AR | 2156 [1860, 2580] | 1/16 | 2178 [1882, 2602] | 5/16 |

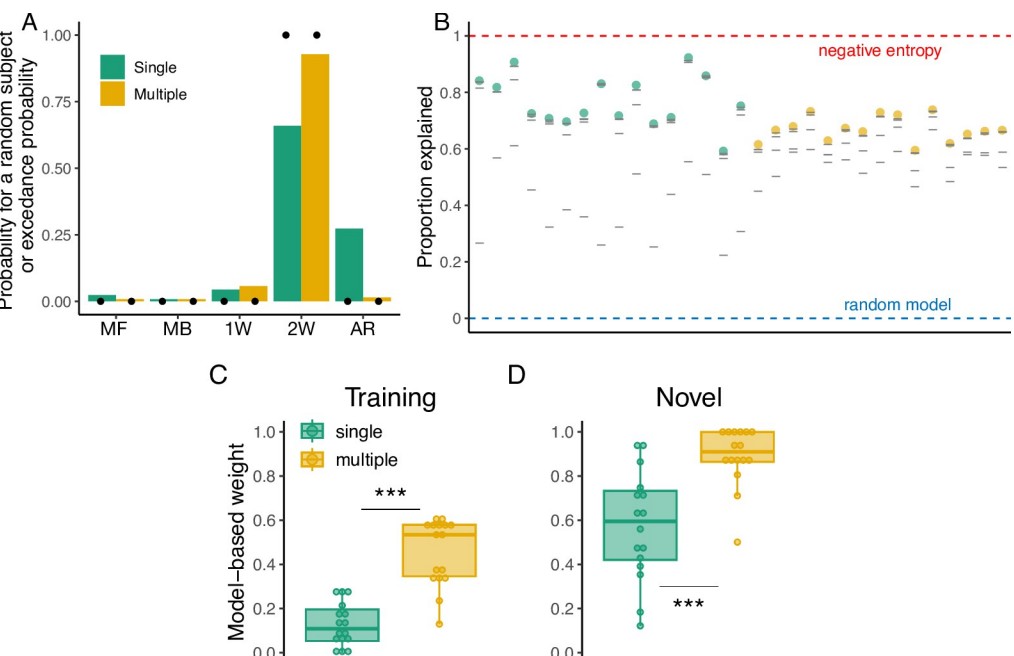

**Fig 8. Modeling results of Experiment 2. (A)** Probability that a random subject taken from the Single and Multiple groups is best described by the tested models (MF: model-free, MB: model-based, 1W: Single-seight hybrid, 2W: Two-weight hybrid, AR: Arbitration model). Black dots indicate the exceedance probability. **(B)** Proportion of the variability in the data explained by the models. Colored dots represent this value for the best model according to the BIC and gray lines represent the other models. Red and blue dashed lines represent the negative entropy (upper boundary) and the performance of a random model (lower boundary), respectively. **(C)** Model-based weight in the hybrid model during training and **(D)** generalization.

significantly higher in the Multiple group during training (adjusted p < 0.001; Fig 8C) and generalization (adjusted p < 0.001; Fig 8D). These results provide further evidence that the observed behavioral differences between the groups during training (Fig 3C) and generalization (Fig 6C) can be described by the components of our model.

When taken together, the behavioral and modeling results from Experiments 1 and 2 indicate that training variability encouraged a robust learning of the visuomotor mapping, which manifested in successful generalization. While the Single group still shows evidence of having learned the mapping, they struggle to generalize even in simple situations that do not require sequential planning. In contrast, the Multiple group maintained ceiling performance regardless of whether planning was necessary (Fig 7). In contrast to Experiment 2, differences in generalization held true when withholding feedback and removing the sequential component in the generalization trials–therefore ruling out the confound of relearning and planning. In addition, those differences remained significant until the end of the generalization phase.

Further supplementary analysis also revealed numerical differences indicating that errors during generalization were more likely to be the result of interference from habitual responses in the Single group than in the Multiple group (S6 Fig). Our behavioral results were supported by our modeling analysis, where a hybrid model with separate weights for the training and generalization phases provided the best description of the data, and significant differences in its weights mirrored the behavioral differences during training and generalization between the Single and Multiple groups (Figs 3C and 6C).

In the following experiment, we inquire whether the benefits in generalization performance due to training variability can be retained over time, even when followed by a long exposure of

repetitive practice with no variability. If this were the case, this would provide evidence that early formation of the visuomotor mapping can reduce the interference of habitual responses in the future and allow for flexible behavior.

## Experiment 3

In the previous experiments, we demonstrated that the Single group's performance consistently decreased during the generalization phase. In this study, we investigated whether a short exposure to training variability could prevent this decline, even when followed by a prolonged period with no variability. Encouraging participants to initially learn the mapping may afford generalization even after a long period of repetitive training at a single target. Alternatively, participants could eventually forget the mapping if they were only required to repeat movements to one target location. This is like if a budding pianist first learned their scales ("mapping") but then practiced only a single melody ("sequence", for an extended period, they might forget the meaning of the piano keys in relation to the scale degrees. With these two points in mind, we designed Experiment 3 to test whether repetitive training to a single target would afford generalization even after a long period of training or it would cause forgetting of (or interference with) the full mapping, while also ensuring that the participants had learned the mapping in the first place.

To test this idea, 16 participants were briefly trained with four start-target pairs (80 trials; now Multiple trials), followed by a first generalization phase (20 trials; Fig 1B). Subsequently, they were exposed to a single start-target pair for a prolonged period of time (1000 trials; now Single trials), followed by a second generalization phase (20 trials). As in Experiment 2, the target locations were placed one move away from the starting point and no feedback was provided. If a visuomotor mapping is trained early on and maintained in memory–even if not being used–then participants would show good performance both in the first and second generalization phase. Alternatively, the prolonged period with no variability in the Single trials, most likely dominated by model-free processes, could impair the previously learned mapping resulting in good performance in the first but not in the second generalization phase.

## Behavioral results

We found that prior to starting each generalization phase, the performance of participants was not statistically different between the Single trials and Multiple trials (t(15) = 1.58, p = 0.13; Fig 9B). Most notably, generalization performance was not significantly different after the Single and Multiple trials (t(15) = 1.05, p = 0.3; Fig 9C). Similarly, there was no significant change in performance from the end of the Multiple trials to the beginning of the first generalization phase (t(15) = 1.05, p = 0.3), nor from the end of the Single trials to the beginning of second generalization phase (t(15) = -1.43, p = 0.17). Furthermore, we did not find a significant difference among the change in performance in the two trial phases (t(29.52) = -1.88, p = 0.07; Fig 9D).

In contrast with the comparable performance of participants in terms of optimal arrivals, their RTs differed considerably among the different stages of the experiment. In particular, RTs were significantly higher at the end of the Multiple trials compared to the end of the Single trials (t(15) = -3.28, p = 0.003; Fig 10B), suggesting that, as in our previous experiments, during the Multiple trials participants were relying more heavily on knowledge about the mapping (i.e., model-based computations). Crucially, we still found an increase in RTs from the end of Single trials to the beginning of the second generalization phase (t(15) = 5.18, p < 0.001; Fig 10D), suggesting a switch from state-action associations to model-based computations. However, in contrast to Experiment 1 and Experiment 2, such increase in RTs was accompanied by

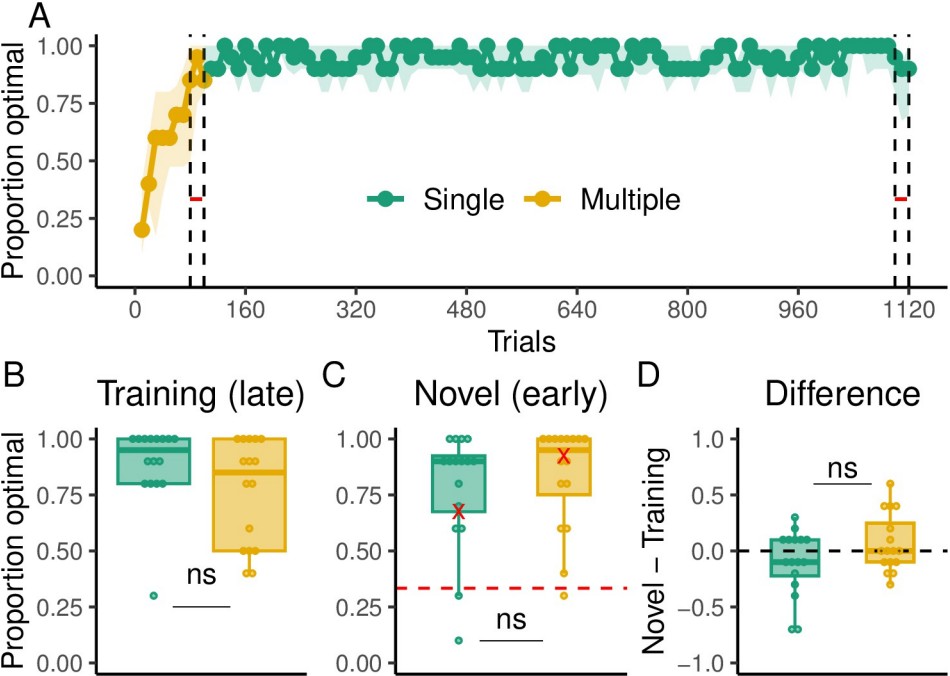

**Fig 9. Behavioral performance in Experiment 3. (A)** Proportion of optimal arrivals over trial bins. Participants were exposed to the Multiple trials (gold) followed by the Single group (green). The solid dotted line represents the median and the shading the interquartile range. The first (trials 81–100) and second (trial 1100–1020) generalization phases are demarcated with the dashed vertical lines. The red dashed line indicates chance level of performance. **(B)** Proportion of optimal arrivals in the last bin of training trials. **(C)** Proportion of optimal arrivals in the first bin of generalization (novel pairs). Red marks indicate performance in the very first trial of generalization for all subjects. **(D)** Difference in the proportion of optimal arrivals between the first bin of generalization and the last one of training.

good generalization performance. As in our previous experiments, we did not observe a change in RTs from the end of the Multiple trials to the beginning of the first generalization phase (t(15) = -1.52, p = 0.14; Fig 10D), which suggest that participants did not have to switch to a different algorithm during this transition. Notably, given the extended practice in the Single trials, all the inter-key-intervals were significantly lower (p < 0.01) than in the Multiple trials (S2 Fig), which was not observed in the previous experiments.

## Modeling results

We evaluated our previous models with the only change that, instead of the 2W hybrid model, we incorporated a hybrid model with four weights (4W hybrid model), one for every stage of the experiment (Multiple trials, first generalization phase, Single trials and second generalization phase). Table 3 shows the results of the individual model comparison. We found that for both the AIC and BIC, the 4W hybrid model proved to be the best model for all participants. This finding was corroborated by our group model comparison, where there was a high probability (>99%) that a random subject taken from our population was best described by the 4W hybrid model (Fig 11A). Similarly, the exceedance probability indicated that the 4W hybrid model was better than the other models with high probability (>99%). Among all participants, this model was able to capture a median of 75% of the explainable variability in the data (Fig 11B).

We compared the model-based weights of the hybrid model across the different experimental stages using a 2-Way ANOVA with experimental phase (training or generalization) and

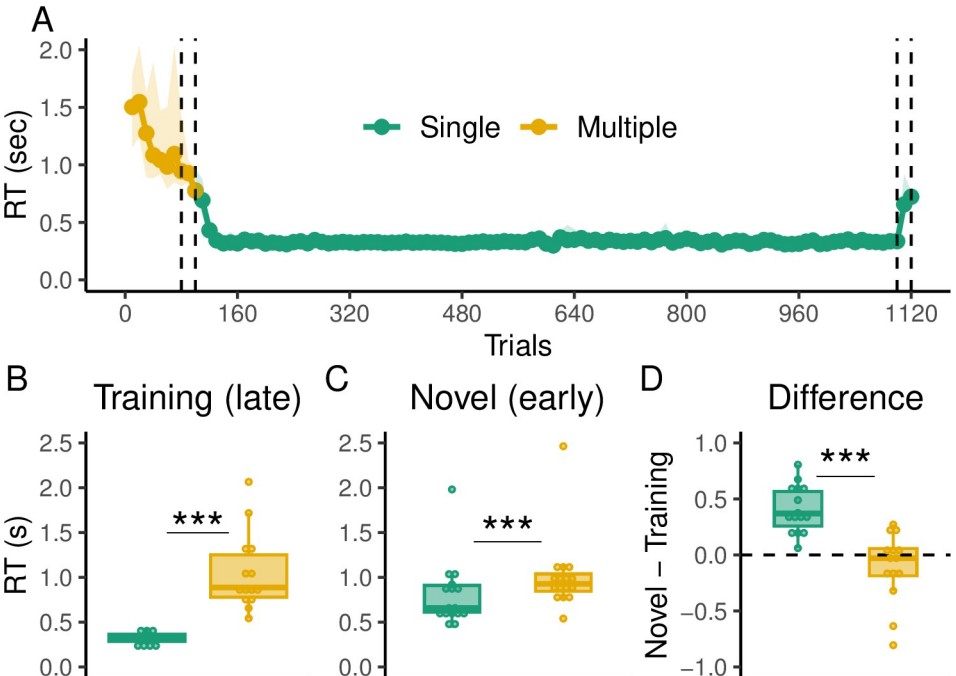

**Fig 10. RTs in Experiment 3. (A)** Per-subject medians of RTs across trial bins (1 bin = 10 trials). Gold data points indicate the Multiple trials and the green data points the Single trials. The first (trials 81–100) and second (trial 1100–1020) generalization phases are demarcated with the dashed vertical lines. The solid dotted line represents the median and the shading the interquartile range. **(B)** RTs in the last bin of training trials. **(C)** RTs in the first bin of generalization (novel pairs). **(D)** Difference in RTs between the first bin of generalization and the last one of training.

trial type (Single or Multiple) as factors, which revealed a significant main effect of the experimental phase ($F(1,60) = 48.96$, $p < 0.001$, $\eta^2 = 0.31$) and trial type ($F(1,60) = 23.36$, $p < 0.001$, $\eta^2 = 0.15$), indicating that the model-based weights were in general higher during generalization and for the Multiple trials. However, post hoc pairwise comparisons revealed that there was no difference between the model-based weights between the generalization phases (adjusted $p = 0.99$), but there was a significant difference during the Single and Multiple trials (adjusted $p < 0.001$), with the model-based weights reaching near zero values for the prolonged Single trials (Fig 11C).

The results from Experiment 3 indicate that, even though participants were most likely relying on state-action associations during the prolonged period with no variability as reflected in low RTs, they were able to flexibly switch to model-based computations when presented with novel start-target pairs, as revealed by increased RTs and successful generalization. Such findings, which were supported by differences in the weights of the 4W hybrid model, show that a

**Table 3. Individual model comparison for Experiment 3.** The AIC and BIC columns show the median across subjects and the interquartile range inside the square brackets. AIC wins and BIC wins columns indicate the number of participants where the given model was the best one according to each metric.

| Models | AIC | AIC wins | BIC | BIC wins |
|---|---|---|---|---|
| MF | 8910 [6991, 10941] | 0/16 | 8924 [7005, 10955] | 0/16 |
| MB | 14562 [12110, 15820] | 0/16 | 14569 [12117, 15827] | 0/16 |
| 1W | 8788 [6980, 10882] | 0/16 | 8816 [7007, 10911] | 0/16 |
| 4W | 8595 [6838, 10750] | 16/16 | 8644 [6887, 10799] | 16/16 |
| AR | 8750 [7159, 10978] | 0/16 | 8778 [7187, 11006] | 0/16 |

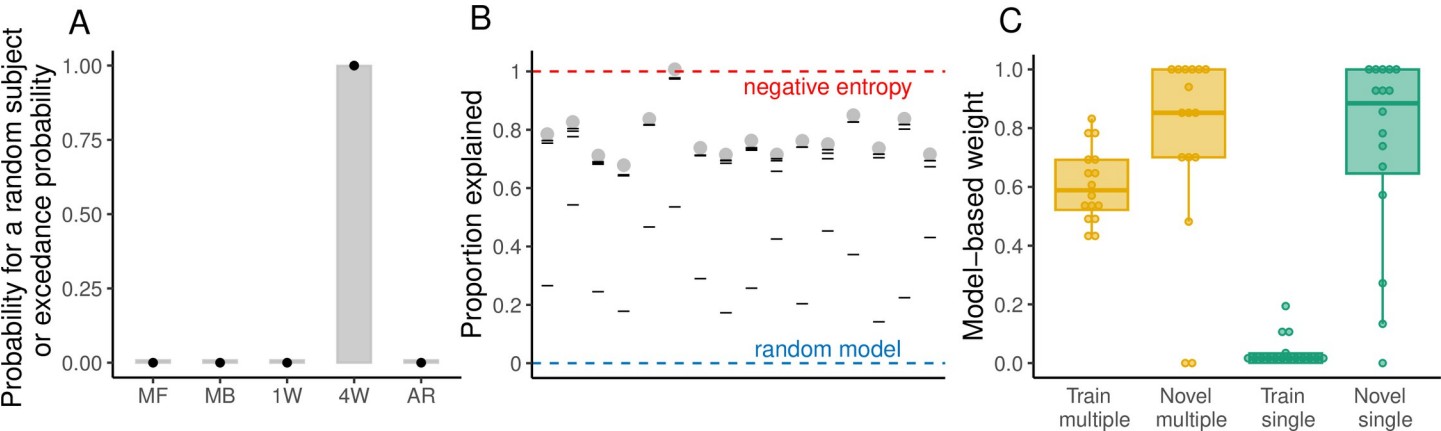

**Fig 11. Modeling results of Experiment 3. (A)** Probability that a random subject is best described by the tested models (MF: model-free, MB: model-based, 1W: Single-seight hybrid, 4W: Four-weight hybrid, AR: Arbitration model). Black dots indicate the exceedance probability. **(B)** Proportion of the variability in the data explained by the models. Gray dots represent this value for the best model according to the BIC and black lines represent the other models. Red and blue dashed lines represent the negative entropy (upper boundary) and the performance of a random model (lower boundary), respectively. **(C)** Model-based weight in the hybrid model during training and generalization in the Single and Multiple trials.

brief exposure to high variability can allow the formation of robust knowledge about the visuo-motor mapping which can lead to successful generalization in the future.

## Experiment 4

It is well-established that skill learning can involve both explicit and implicit processes to varying degrees depending on the training conditions [57–59]. In our prior experiments, the visuomotor mapping was relatively simple and deterministic (i.e., three keys map to three cursor directions) and, as a result, it is possible that participants may have developed explicit knowledge about it and/or the sequence of key responses. If successful generalization was dependent on an explicit representation of the mapping, one potential scenario is that generalization performance will be impaired, particularly in the Multiple group, when learning occurs implicitly. To address this possibility, we introduce a stochastic grid navigation task for the Single (n = 16) and Multiple (n = 16) groups, where on every move there was a probability of 0.2 that the key press could move the cursor to any of the adjacent locations different from the original mapping (see *Materials and Methods* for details). Introducing stochasticity between stimulus-response mappings is a common method to blunt awareness and explicit learning in studies of motor sequence learning [57,60]. In addition, we evaluated the level of explicit knowledge of the visuomotor mapping at the end of the task by asking participants where they thought the keys move the cursor to.

While adding stochasticity could impair the generalization performance in the Multiple group if the mapping was learned explicitly in our previous experiments, it is unclear the extent to which such variability in the key-outcome relationship would affect the performance in the Single group. One possible scenario is that, as in the Multiple group, any benefit from the explicit use of the mapping would go away, therefore making their generalization performance even worse. Alternatively, introducing key-outcome variability would, by default, prevent the repetitive sequence generation behavior that a deterministic mapping would induce, which could result in an enhanced use of the mapping.

Therefore, by leveraging this method, we ask two main questions: 1) Is better generalization performance observed in the Multiple condition, the result of an explicit representation of the visuomotor mapping? 2) Will the training variability induced by a stochastic mapping lead to better generalization in the Single group?

## Behavioral results

As a result of the stochasticity in the task, optimal arrivals were rare, therefore as a behavioral measure of performance we considered only arrivals to the target, regardless of the number of key presses. As in Experiments 1 and 2, the Multiple group learned more slowly than the Single group as revealed by a mixed effect analysis and where the Single group was significantly better than the Multiple group in 20 out of the 26 trial bins of training (p < 0.05; Fig 12A). However, asymptotic performance was similar between the groups by the end of the training phase (t (26.73) = 1.31, p = 0.19; Fig 12B). More importantly, there were no differences in generalization performance (t 27.74) = -1.3351, p = 0.1927; Fig 12C) or in the change of performance from training to generalization (t (28.32) = -1.88, p = 0.07; Fig 12D). However, the Single group did significantly decrease its performance from training to generalization phases (t(15) = -2.65, p = 0.01) whereas the Multiple group did not (t(15) = -0.34, p = 0.73).

In contrast with the comparable performance in terms of target arrivals, we found that RTs at the end of the training phase remained significantly higher in the Multiple group (t(28.65) = -2.28, p = 0.02; Fig 12F). However, no difference in RTs was found between the groups at the beginning of the generalization phase (t(26.6) = 1.28, p = 0.21; Fig 12G) as the Single group significantly increased their RTs (t(15) = 11.05, p < 0.001; Fig 12H) to the level of the Multiple group. In contrast, no increase in RTs was found in the Multiple group (t(15) = 0.16, p = 0.87). Similarly, the increase in RTs was significantly greater in the Single group (t(29.36) = 2.74, p = 0.009; Fig 12H).

At the end of the experiment, participants performed an explicit test where they were asked to report the direction the key moved the cursor to. Both groups showed similar degrees of explicit knowledge of the key-to-direction mapping (Fig 13A). For the Single group 68% (11

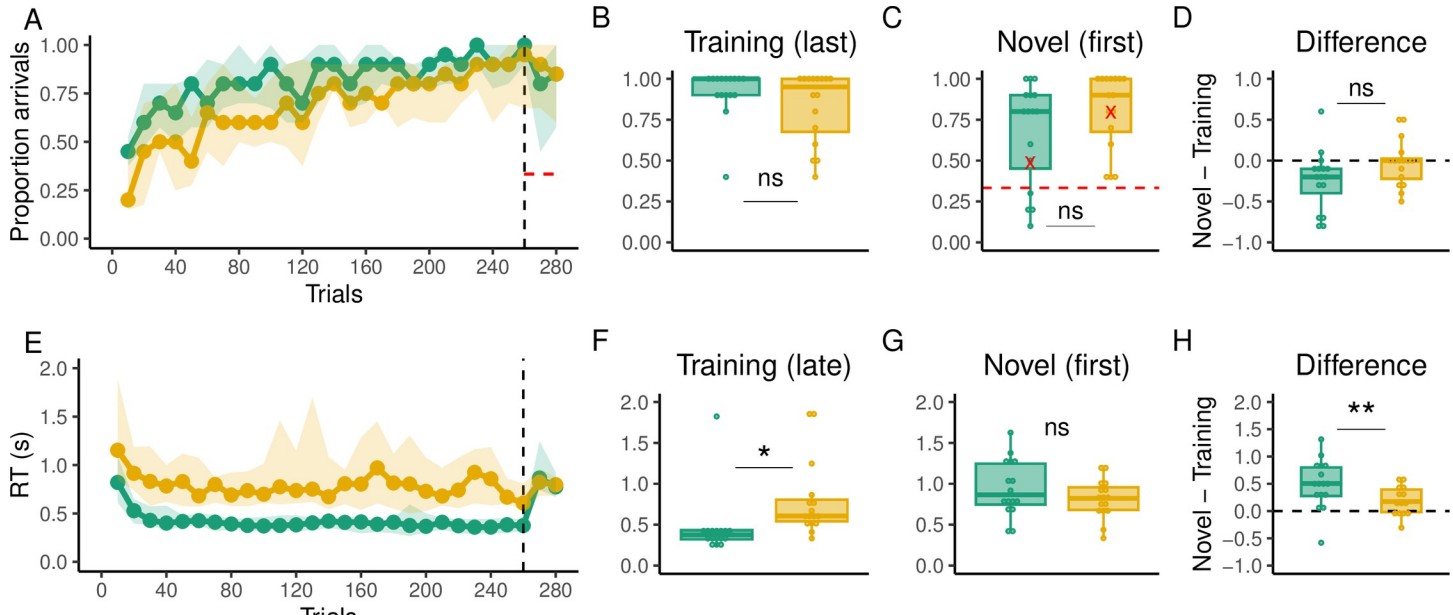

**Fig 12. Behavioral results of Experiment 4. (A)** Proportion of optimal arrivals over trial bins for the Single (green) and Multiple (gold) groups. The black dashed line indicates the beginning of the generalization phase. The solid dotted line represents the median and the shading region the interquartile range. The red dashed line indicates chance level of performance. **(B)** Proportion of optimal arrivals in the last bin of training trials. **(C)** Proportion of optimal arrivals in the first bin of generalization (novel pairs). Red marks indicate performance in the very first trial of generalization for all subjects. **(D)** Difference in the proportion of optimal arrivals between the first bin of generalization and the last one of training. The dashed line here indicates no performance change from training to generalization **(E)** RTs over trial bins. **(F)** RTs in the last bin of training trials. **(G)** RTs in the first bin of generalization **(H)** Difference in RTs between the first bin of generalization and the last one of training.

out of 16) knew all the keys correctly whereas 32% knew two, one or zero key directions. For the Multiple group 62% (10 out of 16) knew all the keys correctly while 38% (6 out of 16) knew two, one or zero key directions. We further explored whether participants that correctly knew the mapping (scoring 3 in the explicit test) in either group had better generalization performance than people that did not know it, or partially knew it (scoring lower than 3). While participants who had full knowledge of the mapping optimally arrived at the target more often on average than participants who had less knowledge, this difference was not significant (t(18.34) = -1.84, p = 0.08; Fig 13B). This suggests that explicit knowledge may not be a strong determinant of how people perform in the task.

## Modeling results

Table 4 shows the results of the model comparison at the individual level. We found that the 2W hybrid model best described the majority of the participants in both groups according to the AIC (12 out of 16 participants in the Single group and 13 out of 16 participants in the Multiple group) and BIC (9 out of 16 participants in the Single group and 8 out of 16 participants in the Multiple group) metrics. In addition, we found that there was a 94% and 76% probability that a random subject taken from the Single and Multiple groups, respectively, was best described by the 2W hybrid model (Fig 14A), and a > 99% probability that such model was better than the others overall according to the exceedance probability. Furthermore, we found that across participants the best performing model captured a median of 81% and 69% of the explainable variability in the data for the Single and Multiple groups, respectively (Fig 14B).

We performed a 2-Way ANOVA over the model-based weights of the 2W hybrid model and found a significant main effect of the experimental phase (F(1,60) = 64.43, p < 0.001, $\eta^2$ = 0.47) and the group (F(1,60) = 10.54, p = 0.001, $\eta^2$ = 0.07), indicating that overall the weights were higher during generalization and for the Multiple group. Nevertheless, when performing post hoc pairwise comparisons, we found that, although there were clear numerical differences between the weights of the groups during training (Fig 14C), this difference did not reach

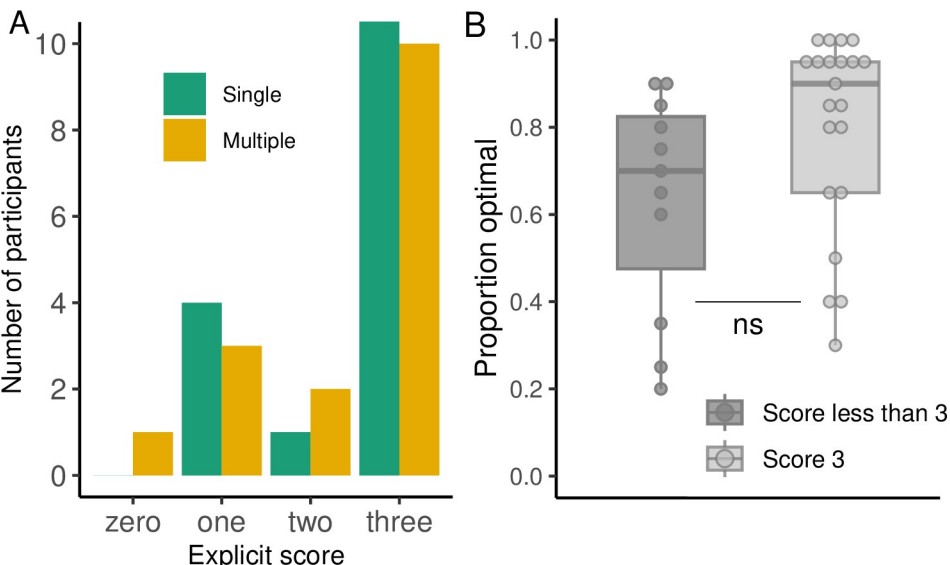

**Fig 13. Results of the explicit test in Experiment 4. (A)** Number of participants that correctly knew zero, one, two or three moving directions of the keys. **(B)** Proportion of optimal arrivals in the generalization phase for participants that score 3 or less than 3 in the explicit test of Experiment 4.

**Table 4. Individual model comparison for Experiment 4.** The AIC and BIC columns show the median across subjects and the interquartile range inside the square brackets. AIC wins and BIC wins columns indicate the number of participants where the given model was the best one according to each metric.

| Groups | Models | AIC | AIC wins | BIC | BIC wins |
|--------|--------|-----|----------|-----|----------|
| Single | MF | 4096 [4297, 6157] | 0/16 | 4702 [4309, 6170] | 5/16 |
|  | MB | 6066 [5458, 7233] | 0/16 | 6072 [5464, 7239] | 0/16 |
|  | 1W | 4627 [4289, 6144] | 3/16 | 4651 [4313, 6168] | 1/16 |
|  | 2W | 4620 [4276, 6137] | 12/16 | 4650 [4306, 6168] | 9/16 |
|  | AR | 4631 [4310, 6144] | 1/16 | 4655 [4334, 6169] | 1/16 |
| Multiple | MF | 7957 [5235, 10349] | 0/16 | 7970 [5247, 10362] | 0/16 |
|  | MB | 8386 [5609, 10658] | 0/16 | 8392 [5615, 10665] | 0/16 |
|  | 1W | 7752 [5032, 10133] | 1/16 | 7778 [5056, 10159] | 3/16 |
|  | 2W | 7736 [5020, 10129] | 13/16 | 7768 [5050, 10162] | 8/16 |
|  | AR | 7786 [5034, 10133] | 2/16 | 7812 [5058, 10159] | 5/16 |

significance levels (adjusted p = 0.07). Similarly, there was no significant difference between the weights of the groups during generalization (adjusted p = 0.15; Fig 14D).

Overall, the results of Experiment 4 show that both groups had similar explicit knowledge about the mapping and that this knowledge was not related to participants' generalization performance. In addition, we found that adding stochasticity to the task improves generalization performance in the Single group to the level of the Multiple group, suggesting that the former learned the mapping, potentially by preventing them from memorizing the sequence solution to the goal. While the Single group was able to generalize to levels comparable to the Multiple group, they did not show the computational demands of the latter during training (also

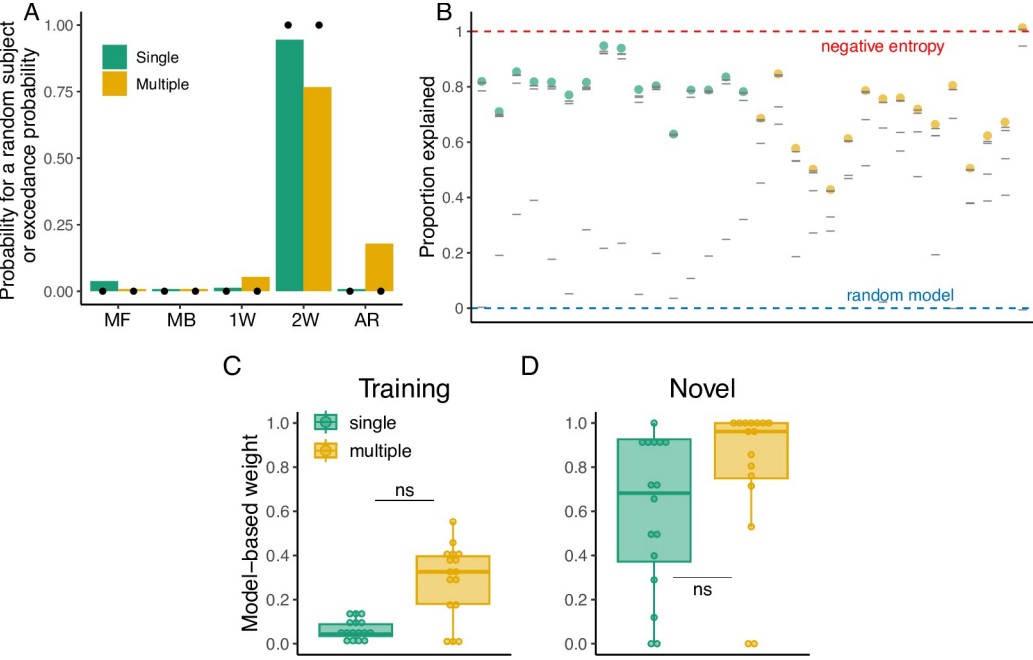

**Fig 14. Modeling results of Experiment 4. A:** Probability that a random subject taken from the Single and Multiple groups is best described by the tested models (MF: model-free, MB: model-based, 1W: Single-seight hybrid, 2W: Two-weight hybrid, AR: Arbitration model). Black dots indicate the exceedance probability. **B:** Proportion of the variability in the data explained by the models. Colored dots represent this value for the best model according to the BIC and gray lines represent the other models. Red and blue dashed lines represent the negative entropy (upper boundary) and the performance of a random model (lower boundary), respectively. **(C)** Model-based weight in the hybrid model during training and **(D)** generalization.

observed in Experiment 1 and 2), as reflected in significantly lower RTs. Therefore, these results suggest that a stochastic training with no variability in the start-target pairs, could bring the generalization benefits of training with multiple start-target pairs, while at the same time reducing its computational cost. Our modeling analyses indicate that such results could arise with stochastic training allowing to appropriately increase the weight of model-based computations during the generalization phase.

## Discussion

A vast number of skills require the formation of novel visuomotor mappings. Sometimes these mappings can be completely arbitrary like in video games, where an "up" press on a video game controller can lead a virtual character to move or jump. The advantage of learning and using these mappings, as opposed to simple state-action associations, is that the mappings can be used for planning and generalization to novel contexts [61–63].

Across four experiments in a grid navigation task, we found that increasing the variability in the number of start-target pairs enables participants to more effectively use the visuomotor mapping and generalize to novel pairs. In particular, the Multiple group showed significantly better generalization performance than the Single group to pairs that required planning (Experiment 1), but also for the ones requiring no planning (Experiment 2). These results suggest that the lack of variability during training can impair the use of the visuomotor mapping even for simple decisions. While the effects of training variability were clear during the generalization phase, we also found evidence that participants in the Multiple group made choices that reflected a greater use of the visuomotor mapping during training (Fig 3). In Experiments 3 and 4, we found that the limited generalization observed in the Single group can be enhanced by a short period of variability introduced early in learning or by incorporating stochasticity into the visuomotor mapping. In addition, our modeling results, along with differences in reaction times, indicate that the Multiple group assigned greater model-based control during the task, while the Single group struggled to switch from a model-free system to model-based computations when encountering novel pairs.

It is well known that training variability leads to better generalization performance in a variety of cognitive domains (see [43] for a review). However, the evidence for such effects in motor learning tasks, particularly for motor adaptation, is mixed [41,64]. For example, in a visuomotor adaptation task, Berniker et al [41] found that two groups of participants trained in workspaces of different sizes, which induced low or high variability in the reaching movements, had no differences in generalization performance on a novel workspace. In line with these results, several studies in motor adaptation suggest that learning proceeds rather locally [65,66], with a very limited capacity to generalize beyond the trained range. In addition, Thoroughman and Taylor [67] found that when the spatial complexity of a force field was increased, which necessarily increases variability, generalization actually narrowed rather than broadened.

On the other hand, Braun et al.c found that participants trained with random rotations, which would average out generating no learning but induce movement variability, were able to adapt faster to subsequent rotations compared to a control group. Their results suggest that training variability can lead to structural learning, i.e., visuomotor rotations, with subsequent learning primarily recalibrating the parameters of the rotation, i.e., the angle. It is important to note, however, that in subsequent work we found that the variability-induced benefit to generalization was the result of explicit strategies and not implicit adaptation [68].

Motor adaptation studies reporting limited generalization would suggest that learning in our task proceeds in a different manner [69,70]. In fact, McDougle et al. [69] found that LSJ, a

patient with severe bilateral hippocampal damage, was able to improve her performance in visuomotor adaptation tasks but was unable to learn a novel key mapping for grid navigation. Such results suggest that successful performance in the grid navigation task depends on brain regions known to be crucial for spatial navigation and planning [71], in contrast to the cerebellar-dependant mechanisms involved in motor adaptation [13,72,73].

It is possible, however, that some form of structural learning about the key sequences could have occurred during the training phase of Experiment 1, similar to the results of Braun et al. [64], which could have resulted in better generalization for the Multiple group. Specifically, in Experiment 1, both the training and generalization phases had start-target pairs that were seven moves apart. However, by design, the Multiple group sampled a larger number of sequences during training, which may have inadvertently exposed them to elements of the trajectories needed to reach the targets in generalization. Although no start-target pair in generalization required the exact key press sequence as those in training, some prototypical movement patterns might have been reused to reach the novel targets. However, this scenario was ruled out in Experiment 2, where targets were only one move away during generalization, yet the Single group still showed significantly lower performance throughout the generalization phase. Therefore, we believe that differences in generalization performance observed between the Single and Multiple group were less likely to occur due to differences in structural learning, but instead due to differences in their ability to flexibly use their internal model (visuomotor mapping).

Notably, in spite of the differences in generalization performance observed in Experiment 1 and 2, the Single group did show evidence of using the visuomotor mapping, albeit not as effectively as the Multiple group. For example, they performed greater than chance when the novel targets were one move away. In addition, when contrasting the generalization performance in Experiments 1 and 2, their RTs increased more for distant targets than for proximal targets, indicating that they were planning [74,75]. This result rules out the possibility that the observed change in RTs occurred solely due to the novelty of the targets, or that participants were just responding randomly. In addition, we also found that the inter-key-intervals during the generalization phase of Experiment 1, were significantly higher in the Single group, strongly suggesting that they had to revisit the visuomotor mapping to plan at intermediate steps of the trajectory. In terms of the modeling results, the Single group also showed model-based weights greater than zero during the training and generalization phases of Experiment 1 and 2, suggesting some influence of model-based computations. Lastly, even in the stochastic version of the task of Experiment 4, most participants in the Single group explicitly knew the moving directions of the keys to some extent, which makes it unlikely that they did not know them in the deterministic, and arguably simpler version of the task from Experiment 1 and 2.

We believe the inability of the Single group to use the visuomotor mapping as flexibly as the Multiple group was, in part, the result of the increased competition from habitual responses developed during the training phase. Indeed, a supplementary analysis from Experiment 1 and 2 revealed that, in the Single group more often than in the Multiple group, the errors during the generalization phase started with the same key as the one of the most frequently used sequence during training (S6 Fig)–which suggest a persistence pattern characteristic of a habit. Our computational modeling analyses support these results as the model-based weights in the 2W hybrid model remained consistently lower overall in the Single group, but most importantly, following the transition from training to generalization, reflecting a greater influence of model-free processes. Relatedly, when presented with stochasticity in the visuomotor mapping (Experiment 4), which in principle prevented participants from repeating the same sequence of key presses during training, the Single group matched the performance of the Multiple group in generalization. This type of training could have disrupted the formation of habitual responses and allow the flexible use of the learned mapping.

Previous research has demonstrated that a key characteristic of habitual responses is their persistence after reward devaluation [76,77]. However, a history of reward might not be necessary for the formation of habits as they can emerge simply through the repetition of actions [78,79,80]. While our models implement reward as visually minimizing the (chessboard) distance to the target, it is unclear the extent to which this is actually rewarding for participants. In addition, the explicit presentation of rewards in our task (emoji faces at the end of the training trials) was relatively sparse. Therefore, it seems more plausible that any habitual response emerging during generalization, resulted from its frequent repetition during the training phase [78,79] (S6 Fig), rather than due to its strengthening from reward. One major limitation of our work is that the current version of our models do not explicitly account for such autocorrelation of responses. Instead, our implementation of the model-free system only predicts chance-level responses in the face of novel start-target pairs. This assumption is not accurate and would likely benefit from the incorporation of a persistence component of previously taken actions [79,81,82]. The exact level at which persistence can occur in our task is an empirical question. For example, it can manifest at the global level if a subject has a tendency to press one key more often than the others throughout the task, regardless of the start-target pair or the current location of the cursor. Alternatively, persistence can occur at a more local level, where responses would be more likely to be repeated on a given grid state, or for a given start-target pair. Allowing the models to capture habitual responses during generalization could improve the model fits and lead to a more pronounced difference in the model-based weights between the Single and Multiple groups.

In addition to the potential interference of habitual responses during generalization, the lower performance in the Single group could also be attributed to the abrupt transition from the training to the generalization phase, which the Multiple group experienced to a lesser extent given the trial-to-trial variability. While this is a plausible scenario in the very first generalization trial—where participants in the Single group did perform at chance level (Fig 6C)—it is unlikely to account for their consistently lower performance throughout the generalization phase. Although we did observe some recovery, the Single group continued to exhibit significantly lower performance than the Multiple group by the end of the generalization phase when we controlled for new learning (Fig 6A). In contrast, studies on change-point detection find that performance typically recovers immediately after an abrupt transition when a controller is well known, at least in simple scenarios [83,84].

Finally, a potential reason as to why the Single group could have exhibited worse generalization performance is because trajectory solutions for the unique start-target pair required the predominant use of the left-up and left-down keys, as compared to the right key. This asymmetry in responses could have led to greater errors associated with the latter key. We addressed this potential confound by analyzing if errors during generalization in the Single group were more frequently associated with the key moving the cursor to the right. We attempted to address this question by examining the generalization trials of Experiment 2 as they require a single move. We found that only 17% of the errors in generalization across all participants were related to this key. Therefore, lower generalization performance could not be attributed to the fewer training samples with the right key in the Single group.

In terms of our computational models, we leveraged a well known distinction in reinforcement learning between model-free and model-based algorithms [45,85]. Although there is evidence that humans may use both systems [48,86], it is less clear the way in which they interact with one another. Similar to previous research [46,48], we assumed this interaction could occur as a simple weighted sum of the output of the systems with fixed weights. However, we also proposed a novel arbitration mechanism based on the encounters with familiar states. While this model did not provide the best fit at the population level, it provided the best fit for

several participants at the individual level for Experiments 1, 2 and 4. It remains to be tested whether other arbitration models (for example, [50,87]) can provide a better fit than our arbitration model or than the fixed weight hybrid models.

Similarly, further variations of the planning algorithm can be incorporated into the models. For example, even though Breadth First Search is suitable to solve deterministic tasks like ours, algorithms that incorporate heuristics, uncertainty or resource rational computations like pruning or truncation, seem more likely to be implemented by the brain [71,88]. We decided to implement Breadth First Search (BFS) given its uniform structure which allows it to explore all paths that lead to the target (even if not immediately obvious). This was particularly important because participants often chose moves that deviated from the target but were optimal, which would have been harder to detect using search algorithms that typically favor moves toward the target (e.g., based on chessboard distance or Euclidean distance heuristics). While our model-free algorithm does tend to prefer such moves, the model-based algorithm aimed to capture the less intuitive moves that likely reflected planning. However, our implementation of BFS assumes that participants can plan with infinite depth, which is an unrealistic assumption. Future variations of this model can include limits in the planning depth known as "lookahead" which can be more biologically plausible [71]. Alternatively, learning algorithms that do not incorporate model-based computations have also been shown to generalize by transferring value across states and actions [89]. However, the fact that reaction times were significantly higher in the Single and Multiple group for distant targets (Experiment 1) compared to proximal targets (Experiment 2) strongly suggests that participants were engaged in planning in our task. Additionally, RTs in all our experiments were consistently higher than the inter-key intervals (S2 Fig), which likely indicates the time participants spent preparing the sequence of key presses to reach the target—i.e., planning.

For Experiment 3, we found that the benefit of having variable training over generalization remained even after a long exposure to no variability. We believe this is a result of the formation of the novel visuomotor mapping at an early stage of learning [1], and the abrupt change in variability could have separated the mapping memory from future updates [90,91], preventing it from being forgotten. Subsequently, during the period of no variability, novel state-action associations could have been formed. If separate memories for the visuomotor mapping and the state-action associations were formed, the latter could have potentially been evoked by reducing the preparation time during generalization [92]. Crucially, unlike in Experiments 1 and 2, the increase in reaction times from the end of the Single trials to the beginning of the second generalization phase was also accompanied by successful generalization. This suggests that participants were able to effectively switch to model-based computations.

The results of Experiment 3 corroborate previous findings outside visuomotor adaptation that indicate that the benefits of variable training occur when variability is introduced early in learning as opposed to later [43], but only when variability is not too high. In our experiments, variable training implied being exposed to four pairs of start-target locations, which were repeated at least 20 times each (early in learning Experiment 3) and up to 70 times (Multiple conditions in Experiments 1, 2 and 4), which we believe provided participants enough familiarization with each of them. Had they experienced more variability, for example by changing the start-target pairs every trial, performance could have been slower and the benefits in generalization could have arrived later.

In Experiment 4, we showed that variability in the form of a stochastic mapping can lead to comparable generalization performance between the Single and Multiple groups. However, the significantly lower RTs during the training phase by the Single group, indicate that the improvements in generalization do not come at the expense of higher RTs during training. Therefore, a stochastic training with a single goal, can prove to be an effective regime that

maintains the benefits of both model-free and model-based computations. In addition, while we were not able to fully remove the awareness of the mapping in the task, we did not find evidence that differences in awareness between participants were related to generalization performance. We believe further investigation is necessary to test the role of explicit processes in the learning of visuomotor mappings as previous research suggests they might be crucial in tasks similar to ours [69,70].

Whereas most of previous studies in sequence learning like SRT, *m* x *n* tasks or discrete sequence production have allowed the study of externally generated sequences specified by the experimenter, there has been a recent interest in sequences that humans generate internally [31,33–35], which, by not being constrained, allow us to explore the planning processes that make humans arrive at given solutions to achieve goals. A model task in this direction has been grid navigation. Our work provides a step in this direction by further providing cognitive models of the processes that might generate these sequences: model-based mapping learning and state-action associations. We believe these types of tasks are good models of a variety of the activities that humans perform in their lives such as playing video-games, musical instruments or sports, where improvisation and self selection of actions is a common feature. However, it is important to note that in contrast to the complexity of the mappings that humans learn in their lives, the mapping that we tested in our studies was relatively simple. Therefore, there is a need to study mappings with greater complexity. In a recent work [93], we address this point by asking participants to navigate in the grid using a mapping based on the less intuitive rule of the 'Knight' from chess. This and similar settings can allow the study of the processes, such as learning and planning, involved in the acquisition of complex skills.

Finally, grid navigation as in the current experiments sits at the intersection of motor learning and spatial navigation where the interaction of procedural and declarative processes likely occurs. For example, previous studies have highlighted that participants with some impairment in declarative knowledge do not perform as well as controls in similar tasks [69]. Therefore, grid navigation could be used as a testbed for how declarative knowledge contributes to the acquisition of a motor skill. At the same time, it rests at the level of complexity where it is still tractable to build relatively simple cognitive models to explain human performance.

## Materials and methods

### Ethics statement

The experiments were approved by the Institutional Review Board (IRB) from Princeton University and all participants provided written informed consent before participating in the experiment.

### Participants

112 undergraduate students (49 males, 58 females, 4 non-binary and 1 preferred not to say; mean age = 19.9, sd = 1.4) from Princeton University were recruited through the Psychology Subject Pool. Sample sizes were based on prior studies of the grid navigation task [31,33,34].

### Apparatus and task design

All experiments were performed in person using the same computer equipment. Stimuli were displayed on a 60 Hz Dell monitor and computed by a Dell OptiPlex 7050'a machine (Dell, Round Rock, Texas) running Windows 10 (Microsoft Co., Redmond, Washington). Participants made their responses using a standard desktop keyboard. All experiments were

programmed in CSS, Javascript and HTML, and run on a web browser and hosted on Google Firebase. Subjects were seated in front of the computer and were asked to follow the instructions to begin the task.

We employed a variant of the grid navigation task based on Fermin et al. [33,34] in which participants were required to navigate a cursor from a starting position to a target location on a 9x9 grid using the J, K and L keys of their keyboard. On Experiment 1–3, each key moved the ship deterministically to one of three possible directions: right, down-left or up-left. On Experiment 4, the keys' directions followed a stochastic rule (see below). At the onset of the experiment, participants were provided with the following instructions *"In this game, you will use the letters J, K and L of your keyboard to move a vehicle through a grid to a target location. Your goal is to arrive using the shortest route. If you arrive with the shortest route, you will see a happy face. If you arrive using a different route, you will see a neutral face. If you do not arrive after a certain time, you will see a sad face."* After participants confirmed they understood the instructions, the task began. The cursor was displayed as a ship and the targets as anchors. Additionally, to make the task more engaging, it was performed with a background of the ocean with quiet wave sounds and 'bubble' sounds every time the cursor moved.

On a given trial, the cursor and a target appeared in locations that varied across experiments (see Fig 1). Depending on the performance of the trial, subjects could receive three types of feedback. If they did not arrive at the target in less than 10s, a sad face appeared in place of the target, along with a "wrong sound" indicating they had failed. If participants arrived at the target but not in the minimum number of key presses, a neutral face and sound were presented. If they arrived using the minimum number of key presses, a happy face with a "correct sound" was presented. The visual feedback remained on the screen for 1s after which an inter trial interval of 500 ms occurred. Then, the next trial began. The experiment was divided into a training and a generalization phase, which will be described in detail for each experiment below. During the training phase of all experiments, the targets were placed seven moves away from the start location.

## Experiment 1 Procedure

The goal of this experiment was to determine whether specific training regimes with different levels of variability promote the formation of local state-action associations or flexible and generalizable visuomotor mappings. For all participants, the J, K and L keys moved the ship to the down-left, right and up-left, respectively (Fig 1A). The training phase consisted of 260 trials which were followed by 20 generalization trials interleaved with 20 training trials, giving a total of 300 trials. We chose the number of trials so the experiment lasted between 40–60 min, which corresponds to one hour of credit for our subjects (undergraduate students). However, we were able to replicate our main findings with an online study where we reduced the number of trials to 100 (80 training and 20 generalization; S10 Fig). Subjects were randomly assigned to one of two groups that differed in the number of start-target pair locations presented during training (Fig 1). In the Single group (n = 16), a unique start-target pair was presented for all training trials. In this case, the target could be reached using a unique sequence of key presses (e.g., J-L-J-L-J-L-K), however, participants were not constrained or encouraged to do so.

For the Multiple group (n = 16), four start-target pairs were presented throughout training, where each of them appeared 65 times. We randomized the pairs such that the same pair did not show up more than twice in a row and all four pairs appeared once before observing them again. Additionally, the target for each pair could not be reached using the same sequence of key presses that arrived at other targets. In the generalization trials, four novel start-target

pairs were presented for both groups. The target was placed seven moves away from the starting point just as in the training trials. Each of the generalization pairs was repeated five times but no pair appeared more than twice in a row, and all four pairs were observed before showing them again. No performance feedback (emoji faces and sounds) was provided in generalization trials, but movement feedback after each move was available. However, movement feedback was provided in both the interleaved training trials and the generalization trials.

## Experiment 2 Procedure

In Experiment 2, we tested whether differences in generalization performance between the Single and Multiple groups in Experiment 1 would remain when movement-related feedback was withheld and no sequential planning was required. To achieve this, we modified the experimental setup by situating the target locations during generalization trials only one step away from the starting point (as opposed to seven in Experiment 1). Moreover, we removed the training trials interleaved in the generalization phase.

The novel start-target pairs for generalization were created by linking four start locations with three possible target locations (Fig 1B). All pairs were presented at least once, and the remaining generalization trials were randomly chosen without replacement from the available twelve. Finally, In order to control for mapping-specific effects, we randomized the directions each key was assigned to across subjects. The training phase was the same as in Experiment 1 for the Single (n = 16) and Multiple (n = 16) groups. The total number of trials was 280.

## Experiment 3 Procedure

In Experiment 3, we tested whether a short exposure to the training variability followed by a long exposure to no variability would be sufficient to learn the mapping as well as to maintain a memory of the mapping, thus affording good performance in the generalization phase at the end of the experiment. Participants (n = 16) first trained during 80 trials with four start-target pairs (Multiple trials), then experienced a first generalization phase of 20 trials with target locations being one move away as in Experiment 2. We chose 80 training trials of training given that in Experiment 1 and 2, asymptotic performance was reached in this time frame by the Multiple group. In addition, in online pilot study, we found that successful generalization was possible with this number of trials in the Multiple group. Following the first generalization phase, participants were exposed to 1000 trials with a single start-target pair (i.e., Single trials), which was the same as in the Single group of Experiment 1 and Experiment 2. The number of trials in this stage was decided based on a pilot study where we first tested 400 Single trials and found no decline in performance in generalization. Therefore, we aimed to extend this phase further to more robustly test the effect of early training with variability. Finally, participants were exposed to a second generalization phase of 20 trials. Importantly, the order of the start-target pairs in the two generalization phases was randomized.

## Experiment 4 Procedure

The goal of Experiment 4 was to twofold. First, we sought to test whether the visuomotor mapping was represented explicitly or implicitly. Second, we tested whether training variability induced by stochasticity in the key-to-direction mapping during training would prevent explicit memorization of the sequence and pressure learning of the mapping, which would afford generalization. To accomplish this, we imposed a probabilistic rule over the movement of the cursor. Specifically, during the training phase of both the Single (n = 16) and Multiple (n = 16) groups, there was a 0.2 probability that, on every move, the key moved the cursor to any of the other seven directions different from the original mapping (left-down, right and

left-up). We use this probability based on previous studies on sequence learning that have found that adding this level of stochasticity prevents participants from learning sequences explicitly [57,60]. In order to evaluate subjects' awareness of the visuomotor mapping of the task, we asked them at the end of the experiment to indicate the direction each key moved the cursor to. Specifically, pictures of the keyboard keys were displayed on the screen (J, K and L), each of them followed by eight moving options indicated with arrows (top, top-right, right, down-right, down, down-left, left and up-left). Participants had to select among the options the one they believed was the true moving direction of the key. Generalization trials were the same as in Experiment 2, with targets being one step away from the start locations and no feedback was provided.

## Behavioral data analysis

All analyses were performed using the R statistical software [94] or Matlab version 2022a [95]. Our main behavioral measure was optimal arrivals and reaction times to the target in Experiments 1–3, which was defined as the minimum number of key presses to move the cursor to the target (7 moves or 1 move, depending on the experiment and phase). In Experiment 4, due to stochasticity, our primary measure was simply arrival to the target even if it was not in the minimum number of key presses. We also examined both reaction time, defined as the time between target presentation and the first key press, and inter-keypress interval, defined as the time between each successive key press. These behavioral metrics were binned every ten trials. When relevant comparisons were done between our Single and Multiple groups, we used Welch's t tests for unequal variances [96]. Paired t tests were used for comparisons in Experiment 3, as samples were dependent. When comparing the learning curves between the Single and Multiple groups in Experiments 1, 2 and 4, we performed a mixed effect model with time and experimental group as fixed effects, and with subjects as random effects.

## Computational modeling

In order to gain mechanistic insight into the learning processes that could have given rise to the results of our experiments, we evaluated five computational models in all our experiments. At one end of the modeling spectrum, we implemented a prediction error RL model to characterize inflexible, habitual behavior, which we believe could be induced in our Single group (model-free). Although this model works in a relatively straightforward manner, it predicts poor generalization as it can only know what to do in situations it has experienced in the past (Fig 4). At the other end of the modeling spectrum, we used a Bayesian model along with a tree-search planning process, to represent a learner that acquires the true key-to-direction mapping and leverages it to decide the best course of action (model-based). As we will describe below, this model would be able to generalize well in our task and we believe a similar mechanism could be drawn on in our Multiple group. Finally we considered three hybrid models that differ in the specification of the mixing weight between the model-free and model-based algorithms. In particular, we considered a hybrid model with a single weight, with two weights (one for the training phase and one for the generalization phase) and with a time varying weight where the arbitration between the systems is based on the familiarity of the current state.

*Model-free (MF)*: This model uses prediction errors to update the value of the keys at each grid cell for every target location using absolute coordinates. Model-free algorithms have received considerable attention in the past years due to its simple trial-and-error mechanism, which can capture a wide variety of behavioral and neural data [20,86,97]. In our task, it updates the value $v$ for pressing key $k$ after a prediction error $\delta$ is observed. More explicitly, for

every time step t:

$$v_t^k = v_{t-1}^k + \alpha \delta_{t-1}^k$$

$$\delta_{t-1}^k = r_{t-1}^k - v_{t-1}^k,$$

where r is the reward obtained and $\alpha$ is a free parameter that modulates the speed of learning. We define reward r in terms of the reduction of the chessboard distance d to the target. Specifically:

$$r = 1 \ if \ d_t < d_{t-1} \ or \ r = 0 \ if \ d_t \geq d_{t-1}$$

$$d = max(x_{target} - x_{cursor}, y_{target} - y_{cursor}),$$

x and y are the grid coordinates of the target and cursor. Then, the probability for pressing key k at time step t is generated using a Softmax function:

$$\phi_t^k = \frac{e^{\beta v_t^k}}{\sum_{k=1}^3 e^{\beta v_t^k}}$$

$$R_t \sim Cat(\phi_t^1, \phi_t^2, \phi_t^3),$$

where $\beta$ is the inverse temperature parameter and $R_t$ is the key press at time step t. This model has two free parameters: $\alpha$ and $\beta$. Note that whereas many model-free approaches (temporal-difference methods, etc.) to multi-step decision tasks of this sort recursively learn a multi-step value function measuring distance to goal, here we streamline this approach slightly by defining the target value at each step non-recursively, in terms of the simple chessboard heuristic at each step. This is similar to advantage learning (itself a variant of the actor-critic), but with the value function component fixed as the chessboard distance. In a supplementary analysis (S11 Fig), we show that this model-free algorithm performs better or at the same level than SARSA, a commonly used temporal-difference algorithm [20]. We believe that the reduction in the chessboard distance is an intuitive measure of reward in this model, as it is equivalent to visually getting closer to the target. In addition, given the relatively few explicit rewards in our task (emoji faces at the end of the trial) compared to the number of moves, incorporating the distance reduction as a reward signal aimed to aid learning in the model to more closely resemble the fast learning curve observed in the data. However, this form of distance assumes the cursor can move to any of the adjacent locations, which is not true in our experiments, but is reasonable in an agent that has no knowledge of the key-outcome mapping. As we will see in our next model, the distance to the target can instead be measured as the number of key presses away from it. When the available moves of the cursor are constrained, the key-press distance can differ from the chessboard distance. More importantly, knowing the key-press distance implies knowledge of the true key-outcome mapping, a fundamental property of our next model.

*Model-based (MB)*: In this model, a probability distribution over the key-outcome mapping is updated using Bayes rule and subsequently used to reduce the number of key presses away from the target. In particular, for every key k, the cursor movement direction x is assumed to be generated by a Categorical distribution:

$$x_k \sim Cat(\theta_1, \ldots, \theta_8)$$

where $(\theta_1, \ldots, \theta_8)$ are the true probabilities that a given key moves the cursor to each of the

eight adjacent locations. These probabilities are unknown but can be inferred using Bayes rule. In order to do that, a prior distribution over $(\theta_1, \ldots, \theta_8)$ has to be specified which represents the initial knowledge of the key-outcome mapping. For reasons of conjugacy, it is convenient to choose a Dirichlet distribution:

$$(\theta_1, \ldots, \theta_8) \sim Dir(1, \ldots, 1).$$

Making the initial parameters equal to 1 gives no preference for any direction *a priori*. While we believe this is a reasonable starting point, the specification of the initial values of the parameters can affect the sensitivity of the model to the data. For example, larger values can make the model less sensitive to data. Then, the posterior belief about the mapping is described by another Dirichlet distribution:

$$(\theta_1^*, \ldots, \theta_8^*) \sim Dir(\alpha_1, \ldots, \alpha_8)$$

$$\alpha_n = 1 + \Sigma_j 1(j = i),$$

where $\Sigma_j 1(j = 1)$ is the number of times the key was observed to go in the ith direction. The expected value of the parameters $(\theta_1^*, \ldots, \theta_8^*)$ can be computed to have a vector of probabilities $\pi$ instead of a vector of random variables:

$$\pi_i = \frac{\alpha_i}{\sum_{i=1}^{8} \alpha_i},$$

$\pi_i$ is the probability that the cursor goes to the ith direction. That is, if a key is pressed, the cursor can end up in the eight adjacent locations with probabilities $\pi$. In model-based reinforcement learning $\pi$ corresponds to the transition probabilities for a given state and action. Our model is a special case of these algorithms for which the transition probabilities are the same for all states. These probabilities are then used to compute the expected distance to the target in the next time step if that key was pressed:

$$E(d) = \sum_{i=1}^{8} d_i \pi_i,$$

where d is the actual distance to the target, that is, the number of key presses away from it. In order to compute d, we used Breadth First Search (BFS) [98]. BFS transforms our grid environment into a graph where each node represents a grid state and nodes are connected among themselves according to the possible transitions in the grid given the visuomotor mapping. BFS is thought to represent the planning process in the model-based algorithm which is known to work well in deterministic environments like ours [71]. What BFS does is to search on the graph created with the grid environment by first visiting the nodes that are one move away from the current location, then it checks if the target is there; if it isn't, then it continues searching in the nodes that are two moves away and so on. It continues this process until it reaches the target. We can use $-E(d)$ to represent the value of pressing a given key. Changing the sign to negative makes lower distances more valuable, then these quantities can be plugged into a Softmax function:

$$\phi_t^k = \frac{e^{-\beta E(d)_k}}{\sum_{k=1}^{3} e^{-\beta E(d)_k}}$$

$$R_t \sim Cat(\phi_t^1, \phi_t^2, \phi_t^3).$$

This model has one free parameter: $\beta$. Importantly, this algorithm has a different reward signal (the actual distance to the target) compared to the model-free algorithm (chessboard distance). While this is not a common assumption in model-free and model-based models, we reasoned that it could capture the idea that the model-free algorithm does not have access to values resulting from using a model of the world (transition probabilities), whereas the model-based algorithm does.

## Hybrid models

As noted previously [46,48–50] we considered the possibility that participants implemented both model-free and model-based computations. We represented this possibility as weighted sums between the outputs of the model-free and model-based algorithms.

## Single-weight model (1W)

In this hybrid model, there is a single weight across the entire experiment. In particular:

$$\phi_t^{Hybrid} = \omega \phi_t^{MB} + (1 - \omega)\phi_t^{MF}$$

$$R_t \sim Cat(\phi_t^{Hybrid\ [1]}, \phi_t^{Hybrid\ [2]}, \phi_t^{Hybrid\ [3]}),$$

where $\omega$ is the weight for the model-based component. Single-weight hybrid models provide a simple way to specify the interaction between the learning systems [46,48], although they could miss important information about the dynamics–if existent at all. This model has four free parameters: $\alpha$, $\beta_{MF}$, $\beta_{MB}$ and one weight parameter $\omega$.

## Two and four-weight models (2W and 4W)

These hybrid models consider the possibility that the model-based weight, $\omega$, is different during the learning and generalization phases. Such variation can incorporate scenarios where the model-based influence increases when the novel targets are experienced. The 2W hybrid model has five free parameters: $\alpha$, $\beta_{MF}$, $\beta_{MB}$, $\omega_{train}$ and $\omega_{gen}$. For Experiment 3, we considered a four-weight hybrid model which has one weight for the Multiple trials, one weight for the Single trials, one weight for the first generalization phase and one weight for the second generalization phase, giving a total of seven free parameters.

## Arbitration model (AR)

Based on preliminary modeling results, where we fitted an unconstrained model using a time series of weights as free parameters (S7 Fig), we considered that the mixing weight, $\omega$, could change over time. While this model is overparameterized and underperforms the rest of the models (S7 Fig) it provides preliminary insights into the dynamics of the weights. This observation aligns with previous research suggesting a transition from model-based to model-free control, which underlies the formation of habits [48]. Accordingly, we propose an arbitration mechanism where the weight of the model-based component varies as a function of the history with familiar and novel states. In particular, we use a Bayesian updating approach, where the model-based weight on every time step $\omega_t^{MB}$ is given by a Beta distribution:

$$\omega_t^{MB} \sim Beta(\alpha_t, \beta_t),$$

where

$$\alpha_t = \alpha_{t-1} + [S_{t-i} \neq S_t \text{ for } i \in \{1, 2, \ldots, \tau\}]$$

$$\beta_t = \beta_{t-1} + [S_{t-i} = S_t \text{ for } i \, \epsilon \, \{1, 2, \ldots, \tau\}]$$

$S$ is the state, encoded as the current location and the target location. In this update rule, the parameter $\beta$ of the Beta distribution increases by one unit if the current state has not been experienced in the past, with the memory window parameter $\tau$ controlling how far into the past to consider. If novel states are continuously being experienced, the distribution of $\omega^{MB}$ will shift towards 1, increasing the control of the model-base system. On the other hand, if the current state has been experienced in the past, the parameter $\beta$ of the Beta distribution increases by one unit. Therefore, If familiar states are continuously being experienced, the distribution of $\omega^{MB}$ will shift toward 0, increasing the control of the model-free system. In order to to mix the output of the model-based and the model-free system at every time step we use the mean of the Beta distribution: $\omega_t = \frac{\alpha_t}{\alpha_t + \beta_t}$

$$\phi_t^{Hybrid} = \omega_t \phi_t^{MB} + (1 - \omega_t)\phi_t^{MF}$$

$$R_t \sim Cat(\phi_{t-1}^{Hybrid\,[1]}, \phi_{t-1}^{Hybrid\,[2]}, \phi_{t-1}^{Hybrid\,[3]}).$$

This arbitration model provides a simple mechanism of how the weight of the model-based system could differ between the Single and Multiple group across the experiment (S8 Fig). In particular, the frequent changes in the start-target pairs for the Multiple group makes it more likely that previously visited states are more distant in the past, potentially being subject to forgetting (as modulated by the $\tau$ parameter). Therefore, resulting in greater model-based weights during training and at the beginning of the generalization phase. In contrast, for the Single group, familiar states are more likely to be found in the recent past due to the lack of changes in the start-target pair. This results in lower model-based weights during the training phase and at the beginning of the generalization phase.

This model has four free parameters: $\alpha$, $\beta_{MF}$, $\beta_{MB}$ and $\tau$

*Model fitting and evaluation*: We used Bayesian Adaptive Direct search [99] implemented in Matlab code to obtain point estimates of the parameters of our models. For each participant and model, we computed the Akaike information Criterion (AIC) [51] and the Bayesian Information Criterion (BIC) [52].

*Absolute goodness of fit*: In addition to computing the AIC and BIC, which allows us to compare models among themselves, we wanted to see how well they described the data in the absolute sense, that is, compared to a theoretical (near) upper boundary for any probabilistic model, at least given particular assumptions about exchangeability. This approximate upper limit is represented by the negative entropy [54–56] and is given by:

$$-H(p(D|M_{true})) = \Sigma_D \, p(M_{true})log \, p(M_{true}),$$

where $p(D|M_{true})$ represents the probability distribution of the data given the true model. The negative entropy is a non-positive quantity and intuitively represents how much we can know about the data from the true generative model. An estimator of the negative entropy that has small error even with few data points is given by Grassberger [54–56]. For our experiments,

this estimator is given by:

$$-\hat{H}(p(D|M_{true})) = \sum_{i=1}^{C} N_i \left( G_{N_i} - \frac{1}{N_i} \left( K_i^{[1]} G_{K_i^{[1]}} + K_i^{[2]} G_{K_i^{[2]}} + K_i^{[3]} G_{K_i^{[3]}} \right) \right),$$

$$\text{with } K_i^{[1]} + K_i^{[2]} + K_i^{[3]} = N_i,$$

where $G_0 = 0$, $G_1 = -\gamma - \log 2$, and $G_2 = 2 - \gamma - \log 2$. $\gamma \approx 0.577215$ is Euler's constant.

For ($n \geq 1$):

$$G_{2n+1} = G_{2n},$$

$$G_{2n+2} = G_{2n} + \frac{2}{2n+1},$$

Thus,

$$n = 2$$

$$G_{2n} = \overbrace{-\gamma - \log 2 + \frac{2}{1} + \frac{3}{2}}^{} + \cdots \frac{2}{2n+1}$$

$$n = 1$$

$C$ represents partitions of the data, e.g., experimental conditions, which in our case equals the number of unique pairs of start and target locations times the number of states in the grid. Therefore, $C$ was not the same in all the experimental groups. In Experiment 1, the number of states in the grid was 81. The number of start and target pairs was five for the Single group and eight for the Multiple group. Therefore, $C = 405$ and $C = 648$ for the Single and Multiple groups, respectively. In Experiment 2 and Experiment 4, the number of grid states was 81 for both groups. The number of unique pairs of start and target locations was thirteen for the Single group and sixteen for the Multiple group. Therefore, $C = 1053$ in the Single group and $C = 1296$ in the Multiple group. Lastly, for Experiment 3 there were seventeen unique pairs of start and target locations across the experiment, therefore $C = 1377$. $N_i$ is the total number of responses in the partition $i$ of the data. $K_i^{[1]}$ is the number of responses to key 1 in the $i$ partition of the data, $K_i^{[2]}$ the number of responses to key 2 in the $i$ partition of the data and $K_i^{[3]}$ the number of responses to key 3 in the $i$ partition of the data. Importantly, this estimator assumes that the distribution of the data given the true model is stationary, which is not necessarily the case of our task as participants' responses can change due to learning. However, given that subjects' performance stabilized relatively quickly as we can see in Figs 2 and 5, we considered it would be a reasonable approximation to an upper boundary of the models' performance.

The negative entropy can be compared with the negative cross-entropy, which intuitively represents how much we can know about the data from an imperfect model (our models). The negative cross entropy is given by:

$$-H(p(D|M_{true}), p(D|M_i)) = \Sigma_D p(D|M_{true}) \log p(M_i),$$

where $p(D|M_i)$ represents the probability distribution of the data given the proposed model $M_i$. The negative cross-entropy is also a non-positive value. An estimator of the negative cross

entropy is the logarithm of the likelihood function evaluated at the maximum likelihood estimates of the parameters [54], which is returned by our parameter estimation method [99]. In order to provide a simple visualization of the absolute goodness of fit, we computed the proportion of the explainable variability in the data that was explained by the models:

$$Proportion\ explained = \frac{H(p(D|M_{true}), p(D|M_i)) + log\ p(D|M_{rand})}{H(p(D|M_{true})) + log\ p(D|M_{rand})},$$

$log\ p(D|M_{rand})$ is the logarithm of the likelihood of the data given a model that assumes all responses are equally likely and represents a lower boundary for all models. In the numerator, we have what *is* explained by a proposed model (as compared to the lower boundary), relative to what *can* be explained (difference between the upper and lower boundary), which is in the denominator.

*Group model comparison*: In addition to comparing the models among themselves at the individual level, we performed group model comparison following Stephan et al. [53]. Based on their work, the probabilities $(q_1, q_2, q_3)$ of our models in the population follow a Dirichlet distribution:

$$(q_1, q_2, q_3) \sim Dir([\alpha_1, \alpha_2, \alpha_3]),$$

the parameters $\alpha = [\alpha_1, \alpha_2, \alpha_3]$ can be estimated by iterating the following algorithm provided by the authors and that we implemented in R code:

$$\alpha = [1, 1, 1]$$

*Until convergence* :

$$u_{nk} = exp\ (log\ p(y|M_{nk}) + \psi(\alpha_k) + \psi(\sum_k \alpha_k))$$

$$\beta_k = \sum_n \frac{u_{nk}}{\sum_k u_{nk}}$$

$$\alpha = \alpha_0 + \beta$$

*end*,

where k is the number of tested models, n the number of subjects and $\psi$ the digamma function. Importantly, this algorithm only requires that we provide the log marginal likelihood which can be approximated as -BIC/2. In order to avoid extremely big numbers from $u_{nk}$, which returned $\infty$ in R, we used the logarithm of the marginal likelihood with base 100. We iterated the algorithm $10^3$ times to provide reliable estimates of $\alpha$. The new parameters of the Dirichlet distribution can be used to compute the probabilities $[r_1, r_3, r_3]$ that a randomly selected subject follows any of the tested models:

$$[r_1, r_2, r_3] = \left[\frac{\alpha_1}{\tau}, \frac{\alpha_2}{\tau}, \frac{\alpha_2}{\tau}\right],\ with\ \tau = \sum_{i=1}^3 \alpha_i$$

Finally, we computed the probability that a given model k is more likely than the others in the population, i.e., the exceedance probability $\varphi_k$:

$$\forall j \in \{1, 2, 3 | j \neq k\},$$

$$\varphi_j = p(q_j > q_k | \alpha),$$

by the Law of Total Probability:

$$\varphi_j = \int_0^\infty p(q_j > q_k | q_j, \alpha) p(q_j | \alpha) dq_j$$

This integral can be approximated numerically by the method provided in Soch and Allefeld [100] implemented in Matlab code.

## Parameter recovery

In order to verify that the parameters from the tested models were identifiable, we generated data from each of the models using 100 random samples of their parameter space; then, we performed maximum likelihood estimation using Bayesian Adaptive Direct Search [99] to attempt to recover the parameters generating the data. Finally, we plotted the simulated versus the fitted parameters and computed the Pearson correlation between them. As can be observed in S12 Fig, we were able to recover the simulated parameters reasonably well in all cases.

## Model recovery

We simulated 100 data sets from each of the models using random samples of the parameter space. Then fitted those data sets with all the models. In S13 Fig we showed the confusion matrix with the proportion of times that each model was able to recover the data generated by the other models the best according to the Bayesian Information Criterion. In general, all the models were able to recover their own data better than the other models above 85% of the times.

## Supporting information

**S1 Fig. Target specific performance for the Multiple group in Experiment 1.** No apparent differences were found in the performance between the different start-target pairs.
(TIF)

**S2 Fig. Reaction times and interkey intervals in the training phase of Experiments 1–4 (from left to right).** The Single and Multiple groups are shown in gold and green, respectively.
(TIF)

**S3 Fig. Reaction times and interkey intervals in the generalization phase of Experiments 1.** The Single and Multiple groups are shown in gold and green, respectively.
(TIF)

**S4 Fig. Most frequently used trajectories across participants in Experiment 1 and Experiment 2.** Above each grid, we show the percentage of participants for which the depicted trajectory was the most frequently used during training. We show trajectories that were preferred by at least two subjects.
(TIF)

**S5 Fig. Cumulative use of the most frequent trajectories during the training phase in Experiment 1 and Experiment 2.** The red line indicates the maximum value for the start-target pairs in the Multiple group (proportion = 0.25), given that each pair only appeared in one quarter of the trials. The different types of gold lines indicate each of the four start-target pairs in the Multiple group.
(TIF)

**S6 Fig. Evidence for the interference of habitual responses during the generalization phase in Experiment 1 and 2.** For a given subject, we computed the number of errors during the generalization phase where the first move (the only move in Experiment 2) matched the first move of the most frequently used sequence during their training phase. Given that participants in the Multiple group did not make many errors, we aggregated these values across subjects and divided them by the total number of generalization errors across all participants. We report this value in the Y axis of the plot and the dashed redline denotes chance level.
(TIF)

**S7 Fig. Model-based weight over trials for the unconstrained model in the Multiple (gold) and Single (green) groups of Experiment 1 and Experiment 2.** The weights represent a time series of free parameters. A value of 1 reflects fully model-based, while a value of 0 reflects fully model-free. The dashed line demarcates the start of the generalization phase. Since this model has a free parameter per trial, it underperforms the rest of the models when complexity is measured as the parameter counts (Experiment 1: Single: ΔBIC = 2445, Multiple: ΔBIC = 2280; Experiment 2: Single: ΔBIC = 2457, Multiple: ΔBIC = 2214. ΔBIC is the BIC difference in medians with the best model from Tables 1 and 2). However, it provides valuable insights into the dynamics of the weights, which we use to build the arbitration model (AR).
(TIF)

**S8 Fig. Simulations of the model-based weight over time in the arbitration model for different values of $\tau$ in the Single and Multiple groups.**
(TIF)

**S9 Fig. Target specific performance for the Multiple group in Experiment 2.** No apparent differences were found in the performance between the different start-target pairs.
(TIF)

**S10 Fig. Replication of Experiment 1 with Amazon Mechanical Turk participants.** The Single group is represented in green and the Multiple group is represented in gold.
(TIF)

**S11 Fig. Model comparison between our model-free algorithm (here MF\*) and SARSA.** SARSA provides a temporal difference update to state-action values for every start-target pair: $Q(s,a) \leftarrow Q(s,a) + \alpha[r + \gamma Q(s',a') - Q(s,a)]$. We evaluated the models in the data of Experiment 1 and Experiment 2 using AIC and BIC differences and testing if they were different from zero using the Wilcoxon signed-rank test. We found that our model performed the same or better than SARSA depending on the metric. In particular, the AIC difference was significantly in favor of our model for the Single group (V = 34, p = 0.04), although there was no difference between the models in the Multiple group (V = 77, p = 0.33). According to BIC, which penalizes more strongly the extra parameter γ in SARSA, our model-free algorithm was significantly better for the Single (V = 0, p < 0.001) and the Multiple group (V = 26, p = 0.01). We found similar results for the data on Experiment 2, with no significant difference in performance between the models for the Single (V = 46, p = 0.13) or the Multiple group (V = 67, p = 0.97) according to AIC, but our model outperformed SARSA for both the Single (V = 31, p = 0.02)

and the Multiple group (V = 26, p = 0.01) according to BIC.
(TIF)

**S12 Fig. Results of parameter recovery.** On the x axis is the simulated parameter and on the y axis the recovered parameter. The parameters for each model are indicated with different colors (MF = green, MB = orange, 1W = purple, 2W = blue, AR = gold): $\alpha$ = learning rate, $\beta$ = inverse temperature, $\tau$ = memory window, $\omega$ = model-based weight, $\omega 1$ = model-based weight in training, $\omega 2$ = model-based weight in generalization. Red lines represent the linear fit to the data and the gray shading the 95% confidence interval. On the top of each plot we show the Pearson correlation between the simulated and recovered parameters as well as its associated p value. https://osf.io/zwqj9.
(TIF)

**S13 Fig. Confusion matrix with model recovery results.** Numbers inside the cells represent the proportion of times that the model in the Y axis best recovered the data generated by the model on the X axis according to BIC. https://osf.io/4xtmv.
(TIF)

## Acknowledgments

We thank the members of the IPA Lab for helpful discussions.

## Author Contributions

**Conceptualization:** Carlos A. Velázquez-Vargas, Jordan A. Taylor.

**Data curation:** Carlos A. Velázquez-Vargas.

**Formal analysis:** Carlos A. Velázquez-Vargas.

**Funding acquisition:** Jordan A. Taylor.

**Investigation:** Carlos A. Velázquez-Vargas.

**Methodology:** Carlos A. Velázquez-Vargas, Nathaniel D. Daw, Jordan A. Taylor.

**Project administration:** Carlos A. Velázquez-Vargas, Jordan A. Taylor.

**Resources:** Jordan A. Taylor.

**Software:** Carlos A. Velázquez-Vargas, Jordan A. Taylor.

**Supervision:** Nathaniel D. Daw, Jordan A. Taylor.

**Validation:** Carlos A. Velázquez-Vargas, Jordan A. Taylor.

**Visualization:** Carlos A. Velázquez-Vargas, Jordan A. Taylor.

**Writing – original draft:** Carlos A. Velázquez-Vargas, Jordan A. Taylor.

**Writing – review & editing:** Carlos A. Velázquez-Vargas, Nathaniel D. Daw, Jordan A. Taylor.

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
