## [Decision Letter · Decision Letter 0]

14 Aug 2024

Dear Velazquez-Vargas,

Thank you very much for submitting your manuscript "The Role of Training Variability for Model-based and Model-free learning of an Arbitrary Visuomotor Mapping" for consideration at PLOS Computational Biology. As with all papers reviewed by the journal, your manuscript was reviewed by members of the editorial board and by several independent reviewers. The reviewers appreciated the attention to an important topic. Based on the reviews, we are likely to accept this manuscript for publication, providing that you modify the manuscript according to the review recommendations.

The reviewers only had a few minor comments, which should be easy to address.

Sincerely,

Ulrik R. Beierholm

Academic Editor

PLOS Computational Biology

Daniele Marinazzo

Section Editor

PLOS Computational Biology

Reviewer's Responses to Questions

**Comments to the Authors:**

Reviewer #1: I am satisfied with the authors' revisions and recommend that the work be published. My last lingering concern with the work is minor -- in the authors' response to my questions about experiment 3, they presented a justification for the experiment that did not make the text. In particular, I found the following paragraph helpful to my understanding of their work, and suggest similar language be included:

We saw two possibilities, which are outlined in the main text on Lines 611-621: 1) forcing participants to initially learn the mapping may afford generalization even after a long period of repetitive training at a single target (the null hypothesis as the reviewer points out) or 2) participants could eventually forget the mapping if they were only required to repeat movements to one target location. This is like if a budding pianist first learned their scales (“mapping”) but then practiced only a single melody (“sequence”, for an extended period, they might forget the meaning of the piano keys in relation to the scale degrees – the senior author has experienced this firsthand. In addition, in Experiments 1 and 2, it was unclear if the participants in the Single group ever learned the mapping since the task didn’t necessarily demand it. Instead, they could simply memorize a sequence of responses and repeat it throughout the entire experiment. With these two points in mind, we designed Experiment 3 to test whether repetitive training to a single target would afford generalization even after a long period of training or it would cause forgetting of (or interference with) the full mapping, while also ensuring that the participants had learned the mapping in the first place.

Reviewer #2: Overall, the rebuttal comments are satisfactory to the most extent. I commend the authors in conducting additional experiments in addressing the comments and clarifying various things where they offer a rebuttal. Here are few suggestions for improvement. I refer to their point numbers for presenting my concerns/suggestions.

1. Point 2: “However, we were able to replicate our findings of Experiment 1 with an online study where we reduced the number of trials to 100 (80 training and 20 generalization).”

A statement to this effect can be incorporated in the manuscript/supplement to highlight that this protocol used a longer training period than what is typically found in the literature.

2. Point 7: The authors misinterpreted the point of this comment. We meant to suggest that SSG group participants are not exposed to the multi-SG condition. Hence, a peak in RT during the generalization period might suggest adaptation to some extent. A potential way to look at this would be to analyze trial-wise comparison of differences in RT during the generalization period.

3. Point 8: The authors do not seem to address this comment – point number 5 refers to a different kind of analysis. We intended to suggest looking at the normalized RT plots (RT divided by the number of moves) here to control for the number of moves.

4. Point 12: While we agree with the authors’ argument that RL models with sparse rewards performed poorly as compared to those with dense rewards, there is a bit of contradiction here in terms of the interpretation. The chessboard distance reward metric used in these models may not exactly correlate with the KM-based distance. The observation made by the authors that sometimes participants ‘seemingly’ move the cursor away from the target in terms of visual distance while still getting closer in terms of the KM distance can be considered as evidence for model-based learning should be interpreted with caution. Moreover, the point that SARSA with sparse rewards performed poorly could be mentioned in the manuscript or supplement.

5. Point 13: The authors can potentially include a candidate model that is a parametric arbitration model. This model would introduce a good baseline comparison for how an arbitration model (that calls for a model-free or model-based evaluation at each time step in a binary sense unlike the hybrid model that employs weighted estimates from the two systems) performs when the weights are not defined using a series of free parameters.

6. Point 14: Given a substantial amount of variance in RT plots as well as an (in our opinion) unnecessarily long training period, we think it is crucial to introduce perseveration and attention-lapse parameters in the modeling. Without introducing these parameters, our interpretation of best-fit parameters may be conditioned on some model mimicry effects wherein the model may try to adjust the parameters in order to capture the attention lapses and perseveration in the behavioral data.

7. Point 15: We are not sure what the authors are trying to suggest with “Given the recursive structure of …. It was not efficient to use Gibbs sampling”. A full posterior estimation should be plausible even with the kind of model definitions used in this manuscript. Additionally, even with the point estimates, the models should be simulated with the best-fit parameters and the simulated data should be compared to the empirical data.

**Have the authors made all data and (if applicable) computational code underlying the findings in their manuscript fully available?**

Reviewer #1: Yes

Reviewer #2: Yes

PLOS authors have the option to publish the peer review history of their article (what does this mean?). If published, this will include your full peer review and any attached files.

Reviewer #1: **Yes: **Nicholas Franklin

Reviewer #2: **Yes: **Raju Surampudi Bapi

Figure Files:

Data Requirements:

Reproducibility:

References:

---

## [Editor Report · Decision Letter 1]

6 Sep 2024

Dear Velazquez-Vargas,

We are pleased to inform you that your manuscript 'The Role of Training Variability for Model-based and Model-free learning of an Arbitrary Visuomotor Mapping' has been provisionally accepted for publication in PLOS Computational Biology.

Best regards,

Ulrik R. Beierholm

Academic Editor

PLOS Computational Biology

Daniele Marinazzo

Section Editor

PLOS Computational Biology

---

## [Editor Report · Acceptance letter]

20 Sep 2024

PCOMPBIOL-D-24-01082R1 

The Role of Training Variability for Model-based and Model-free learning of an Arbitrary Visuomotor Mapping

Dear Dr Velázquez-Vargas,

I am pleased to inform you that your manuscript has been formally accepted for publication in PLOS Computational Biology. Your manuscript is now with our production department and you will be notified of the publication date in due course.

With kind regards,

Jazmin Toth
